# MeshTok: Efficient Multi-Scale Tokenization for Scalable PDE Transformers

Yanshun Zhao [* 1]   Xiaoyu Peng [* 1]   Jiamin Jiang [2]   Congcong Zhu [2]   Jingrun Chen [1 2]

## Abstract

Conventional patchified Transformers operate on uniform spatial partitions, distributing computational effort evenly across the domain irrespective of local features. This inflexible tokenization scheme is inherently limited in its ability to efficiently represent and process solutions to complex PDEs. To address this, we propose MeshTok, an adaptive mesh refinement (AMR)-inspired tokenization and sequence modeling framework. This method selectively refines spatial regions exhibiting sharp gradients, transient features, or multiscale structures, generating a heterogeneous set of multiscale tokens defined on a fixed simulation grid. These tokens are processed within a unified Transformer sequence, enabling the model to simultaneously capture coarse-grained global context and fine-grained local details without requiring specialized architectural components. Although adaptive refinement moderately increases token count, it promotes a more targeted allocation of computational resources to physically informative regions, which we view as a practical inductive bias rather than a formal optimality guarantee. Experimental evaluations across multiple PDE families and benchmark datasets demonstrate that MeshTok consistently improves the efficiency-accuracy trade-off compared to uniform-grid baselines. This suggests adaptive multiscale tokenization as a scalable and generalizable design principle for neural PDE modeling. Code is available at https://github.com/SCAILab-USTC/MeshTok.

*Equal contribution [1]School of Mathematical Sciences, University of Science and Technology of China, Hefei 230026, China [2]Suzhou Institute for Advanced Research, University of Science and Technology of China, Suzhou 215123, China. Correspondence to: Congcong Zhu <cczly@ustc.edu.cn>, Jingrun Chen <jingrunchen@ustc.edu.cn>.

*Proceedings of the 43rd International Conference on Machine Learning*, Seoul, South Korea. PMLR 306, 2026. Copyright 2026 by the author(s).

## 1. Introduction

Solving partial differential equations (PDEs) is a central yet challenging problem across science and engineering. Classical numerical solvers are highly accurate, but often incur substantial computational cost, particularly for high-resolution simulations or repeated solves across parameterized PDE families (Guo et al., 2016; Tompson et al., 2017). Motivated by the expressive power of deep learning and its fast inference once trained, people have witnessed growing interest in learning-based PDE solvers (Lagaris et al., 1998; Zhu et al., 2025; Li et al., 2023; Brandstetter et al., 2022; Cao, 2021; Li et al., 2020). Especially, physics-informed neural networks (PINNs) (Raissi et al., 2019; Sirignano & Spiliopoulos, 2018; Raissi et al., 2017; Karniadakis et al., 2021; Zhu et al., 2019) and neural operators (Sirignano & Spiliopoulos, 2018; Han et al., 2018) have demonstrated the potential to amortize solution cost and enable rapid approximation for broad classes of PDEs.

Despite this progress, building models that reliably transfer across heterogeneous equations, geometries, parameter regimes, and forcing conditions remains challenging. In practice, learned solvers can be sensitive to distribution shift, and adapting a pretrained model to new settings may still require additional training or calibration. These observations motivate the development of *PDE foundation models*: scalable architectures that can leverage diverse training data and provide strong performance across a wide range of PDE systems under a unified modeling interface.

Transformer architectures provide global interactions, flexible conditioning mechanisms, and strong extensibility, making them promising candidates for scalable PDE modeling. However, directly adopting a ViT-style architecture faces a fundamental efficiency bottleneck. Standard Transformers operate on uniformly partitioned patches, requiring dense tokenization across the entire spatial domain (Vaswani et al., 2017; Rao et al., 2021; Liu et al., 2021). Even when PDE data are defined on a fixed high-resolution grid, the underlying solutions exhibit highly non-uniform complexity: many regions remain smooth and are well captured with low sampling density, while others contain sharp gradients, discontinuities, or multiscale structures that demand finer local representation. Refining all patches everywhere is computationally prohibitive as attention scales quadratically

with the token count, whereas using a single uniform coarse patchification can lead to large errors in challenging regions.

To address this challenge, we draw inspiration from adaptive mesh refinement (AMR) in numerical analysis and introduce a multi-scale tokenization framework that allocates token density where it matters most (Berger & Oliger, 1984; Berger & Colella, 1989; Bar-Sinai et al., 2019; Dzanic et al., 2024; Gillette et al., 2024). Instead of changing the underlying simulation grid, we adaptively vary the tokenization stride on the same high-resolution field: smooth regions are encoded by coarse tokens, while regions with sharp transients or rich multiscale dynamics are assigned fine-grained tokens to capture intricate local behavior. These heterogeneous tokens are jointly processed as a single sequence within a standard Transformer, enabling unified reasoning across multiple spatial scales while maintaining compatibility with existing Transformer architectures. We further design a geometry-aware positional encoding tailored to irregular and multi-scale token layouts by encoding continuous spatial coordinates together with resolution (depth) information, which stabilizes optimization and supports effective cross-resolution interactions. While the adaptive representation introduces controlled computational overhead, it consistently improves accuracy under comparable token budgets and naturally exploits the spatiotemporal coupling structure of PDEs, where local refinements can propagate to global solution quality.

**Our contributions are summarized as follows:**

- We propose a unified AMR-inspired multi-scale tokenization framework built on Transformers for scalable PDE modeling, together with a geometry-aware positional encoding that enables stable training on irregular multi-scale token sets.

- We propose an activity-based indicator to guide token refinement in PDE Transformers, providing an analytically motivated and computationally efficient mechanism for adaptive resolution.

- We provide a theoretical discussion of the proposed AMR-inspired tokenization scheme, offering insights into how adaptive refinement may improve representation efficiency for PDE solutions with localized structures.

## 2. Related Work

### 2.1. Adaptive Mesh Refinement (AMR)

Traditional adaptive mesh refinement methods dynamically adjust spatial resolution during time integration by estimating local error indicators, marking cells for refinement, and refining or coarsening the mesh around salient features (Berger & Oliger, 1984; Berger & Colella, 1989). These approaches allocate computational effort to regions requiring higher fidelity. Recently, machine learning has been explored to automate AMR strategies. For example, Foucart et al. (Foucart et al., 2023) formulate AMR as a deep reinforcement learning problem, training policy networks to guide grid refinement from simulation data while reducing reliance on hand-designed heuristics. Freymuth et al. (Freymuth et al., 2023) propose Adaptive Swarm Mesh Refinement (ASMR), modeling the mesh as a collection of collaborating agents and using graph neural networks to propagate information between neighboring cells. These learning-based AMR approaches have shown promising transferability across related PDE settings and can achieve competitive trade-offs between accuracy and cost compared to classical heuristic refinement strategies in reported experiments. More recently, Xu et al. (Xu et al., 2025) introduce the AMR-Transformer, which integrates an AMR scheme with a transformer-based neural CFD solver to capture long-range interactions in fluid simulation. By combining task-aware pruning with adaptive refinement, AMR-Transformer dynamically adjusts the mesh during inference. Overall, these works reflect a trend toward data-driven refinement frameworks that complement classical error-based heuristics. In contrast, our work adopts AMR as inspiration for *token-level* adaptivity on fixed grids, allocating token density to locally complex regions while retaining compatibility with standard Transformer backbones.

### 2.2. PDE Foundation Models

Neural operator learning has emerged as an effective paradigm for PDE modeling by training networks that map input functions (e.g., initial or boundary conditions) to solution fields. Notable examples include the Deep Operator Network (DeepONet) (Lu et al., 2019) and the Fourier Neural Operator (FNO) (Li et al., 2021), which have demonstrated strong performance on fluid dynamics and weather prediction tasks. These models are often trained in a task-specific manner, and their performance may degrade under distribution shifts such as changes in equations or parameter regimes. To mitigate limited coverage within individual tasks and leverage heterogeneous data sources, recent work explores large-scale pretraining of neural operators across diverse PDE datasets. Zhou et al. (Zhou et al., 2024) systematically evaluate various pretraining strategies for neural operators and find that transfer learning or physics-informed pretraining can improve downstream performance. In parallel, there is growing interest in constructing PDE foundation models via multi-physics pretraining. For instance, Liu et al. (Liu et al., 2024) propose PROSE-FD, which is pretrained on six families of parametrized fluid equations and demonstrates encouraging zero-shot transfer on heterogeneous two-dimensional flow problems. Similarly, Sun et

al. (Sun et al., 2025) explore multi-operator learning and extrapolation as steps toward more general-purpose PDE modeling. Wang et al. (Wang et al., 2025a) introduce the MoE Pre-training Operator Transformer (MoE-POT), which uses a layer-wise gating network to select specialized expert subnetworks. Together, these works suggest a growing trend toward pretraining large neural operators on diverse physics, forming models that can be adapted efficiently and, in some cases, transferred with minimal task-specific tuning.

## 3. Methodology

Conventional Transformer architectures process structured data by partitioning the input domain into uniformly sized patches, allocating computation uniformly across the spatial domain. While this design is effective for natural images or language tokens, it is often suboptimal for partial differential equation (PDE) data. PDE solutions typically exhibit strong spatial heterogeneity: large regions may remain smooth and low-frequency, whereas localized areas contain sharp gradients, discontinuities, or multiscale structures. Uniform patchification can therefore lead to inefficient use of the token budget, over-representing smooth regions while under-representing locally complex dynamics (Roohi & Mahdavi, 2025; Nekoozadeh et al., 2023).

Motivated by this observation, we propose **MeshTok**, an AMR-inspired multi-scale tokenization framework for Transformer-based PDE modeling. As illustrated in Fig. 1, the overall pipeline consists of two main components: an *indicator* and a Transformer backbone. Instead of changing the underlying simulation grid, MeshTok adaptively varies the patch sizes (i.e., tokenization stride) on a fixed-resolution field according to local solution complexity.

The resulting heterogeneous tokens, spanning multiple spatial scales, are processed by a shared Transformer backbone within a unified sequence representation, enabling information exchange across scales. To encode spatial structure under non-uniform patch sizes, we introduce a geometry-aware positional encoding that accounts for both token coordinates and scale (depth) information. This design improves training stability and facilitates effective cross-scale interactions under heterogeneous token layouts. Together, these components allow MeshTok to adapt its representational granularity to the intrinsic complexity of PDE solutions, improving accuracy under comparable computational budgets.

### 3.1. Indicator for Adaptive Patch Refinement

An example of the resulting activity-based refinement pattern is shown in Fig. 2. Under a refinement ratio $k \in (0, 1)$, MeshTok refines the top-$k$ fraction of coarse patches and splits each selected patch into a $2 \times 2$ sub-grid to obtain fine-grained tokens. Unless otherwise specified, we set

$k = 0.25$, which provides a practical balance between refinement effectiveness and computational overhead: smaller ratios may under-refine complex regions, while larger ratios can diminish efficiency gains.

To decide which regions to refine, we use an *activity-based* indicator computed directly from the input field. The key motivation is that PDE solutions are spatially heterogeneous: most regions are smooth and can be represented coarsely, whereas sharp transients and multiscale structures require higher token density. We therefore assign each coarse patch an activity score that measures local variation. Specifically, we compute a per-grid-point activity map by combining gradient magnitude and Laplacian energy,

$$a(u, v) = \|\nabla x(u, v)\|_2 + \lambda \|\Delta x(u, v)\|_2^2,$$

where $\nabla$ and $\Delta$ are implemented using standard finite-difference stencils, and the norm is taken over the channels. This choice is motivated by complementary sensitivities: the gradient term highlights sharp interfaces and shocks, while the Laplacian term emphasizes high-curvature or rapidly oscillating patterns that often correspond to fine-scale structures. We then average $a(u, v)$ within each coarse patch $i$ to obtain its patch score

$$s_i = \frac{1}{|\Omega_i|} \sum_{(u,v) \in \Omega_i} a(u, v),$$

where $\Omega_i$ denotes the set of grid points covered by patch $i$. Finally, we select the top-$\lfloor kN \rfloor$ patches with the largest $s_i$ for refinement, producing a heterogeneous multi-scale token set that is processed by a shared Transformer backbone.

We also consider other refinement indicators, including random refinement, an a posteriori error-improvement criterion, and end-to-end learned indicators; these variants are evaluated in the ablation study (Appendix F). During autoregressive rollout, the refinement scores are recomputed at every prediction step from the model's current input state. After the observed history is exhausted, this state contains the model's own previous predictions rather than ground-truth future labels, so the refinement policy remains fully inference-available. The refinement ratio $k$ is fixed throughout rollout, keeping the token count and compute budget constant even though the selected patch locations may change over time.

### 3.2. Transformer Backbone with Multi-Scale Tokens

**Overview** Our model uses a shared Transformer backbone operating on a unified multi-scale token sequence. Multi-scale here refers to different *patch sizes* (token density) on a fixed $H \times W$ simulation grid, rather than changing the underlying spatial resolution. For temporal modeling, we adopt block-causal self-attention along the time dimension

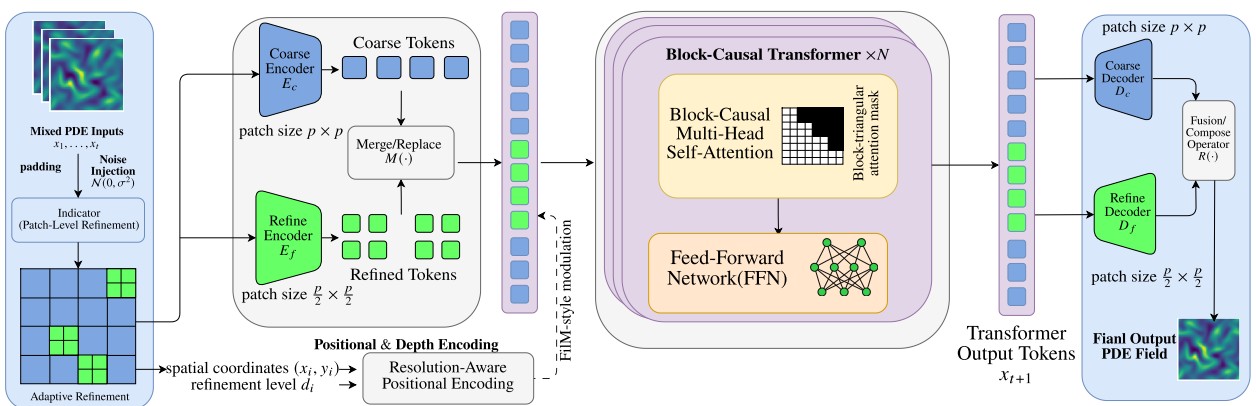

*Figure 1.* An illustration of our model architecture. Given input PDE states, an indicator predicts patch-level refinement scores on a coarse grid. Selected patches are recursively refined to generate a set of multi-scale tokens, which are encoded and merged into a unified sequence with geometry-aware positional encodings. A Transformer backbone processes the resulting tokens to model spatiotemporal dependencies, and the outputs are decoded and fused to reconstruct the PDE solution at the next time step.

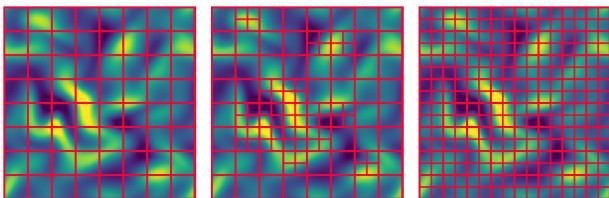

*Figure 2.* Visualization of tokenizations for a $128 \times 128$ PDE field. Left: uniform $8 \times 8$ patch grid (patch size $16 \times 16$). Middle: MeshTok refinement that further splits a subset of coarse patches (25% in this example), concentrating refined tokens near high-variation regions. Right: uniform $16 \times 16$ patch grid (patch size $8 \times 8$), shown with major ($8 \times 8$) and minor ($16 \times 16$) grid lines.

(similar to BCAT (Liu et al., 2025)), such that tokens at time step $t$ cannot attend to future steps $t' > t$, while preserving full attention among spatial tokens within each time step.

**Coarse and Fine Token Embeddings** Given a PDE field $x^t \in \mathbb{R}^{H \times W \times C}$ at time step $t$, we construct patch embeddings at two patch sizes. The coarse encoder $\mathcal{E}_c$ operates on non-overlapping patches of size $p \times p$ and produces

$$\mathbf{Z}_c^t = \mathcal{E}_c(x^t) \in \mathbb{R}^{N_c \times d}, \qquad N_c = \frac{H}{p} \cdot \frac{W}{p}.$$

In parallel, the fine encoder $\mathcal{E}_f$ operates on smaller patches of size $(p/2) \times (p/2)$, yielding

$$\mathbf{Z}_f^t = \mathcal{E}_f(x^t) \in \mathbb{R}^{(4N_c) \times d}.$$

**Indicator-Guided Token Merging** An indicator outputs the indices of coarse patches selected for refinement,

$$\mathcal{I}^t = \text{Indicator}(x^t), \qquad |\mathcal{I}^t| = m, \ \ m = \lfloor kN_c \rfloor,$$

where $k \in (0, 1)$ is the refinement ratio. For each $i \in \mathcal{I}^t$, the corresponding coarse token $\mathbf{Z}_c^{t,(i)}$ is replaced by its four

associated fine tokens extracted from $\mathbf{Z}_f^t$. We denote this operation by a merge operator

$$\mathbf{Z}^t = \mathcal{M}\big(\mathbf{Z}_c^t, \mathbf{Z}_f^t, \mathcal{I}^t\big) \in \mathbb{R}^{(N_c+3m) \times d},$$

which yields a unified multi-scale token sequence at time step $t$.

**Block-Causal Temporal Transformer** Given an input history $\{x^1, \dots, x^T\}$, we apply the above procedure to obtain multi-scale tokens $\{\mathbf{Z}^1, \dots, \mathbf{Z}^T\}$ and concatenate them into

$$\mathbf{Z}^{1:T} = \big[\mathbf{Z}^1, \mathbf{Z}^2, \dots, \mathbf{Z}^T\big].$$

The Transformer backbone processes this sequence with a block-causal attention mask,

$$\widetilde{\mathbf{Z}}^{1:T} = \mathcal{T}_{\text{BC}}\big(\mathbf{Z}^{1:T}\big), \qquad \text{Mask}(t, t') = 0 \text{ for } t' > t,$$

while allowing full attention among spatial tokens within each $\mathbf{Z}^t$.

**Unmerging and Resolution-Consistent Decoding** After global modeling, we use the indicator indices to recover coarse and fine token subsets from the final-step output tokens,

$$\big(\widetilde{\mathbf{Z}}_c^T, \widetilde{\mathbf{Z}}_f^T\big) = \mathcal{S}\big(\widetilde{\mathbf{Z}}^T, \mathcal{I}^T\big),$$

where $\mathcal{S}$ inverts $\mathcal{M}$ by routing tokens back to their coarse or refined groups according to $\mathcal{I}^T$. We then decode both subsets into *the same* $H \times W$ resolution. Concretely, the coarse decoder $\mathcal{D}_c$ maps each coarse token to a $p \times p$ output patch, producing a full-resolution coarse prediction

$$\hat{x}_c^{T+1} = \mathcal{D}_c(\widetilde{\mathbf{Z}}_c^T) \in \mathbb{R}^{H \times W \times C}.$$

Similarly, the fine decoder $\mathcal{D}_f$ maps each refined fine token to a $(p/2) \times (p/2)$ output patch. For each refined coarse

region $i \in \mathcal{I}^T$, the four decoded fine patches tile the same $p \times p$ spatial area covered by the original coarse patch. By assembling all refined regions and leaving unrefined regions empty, we obtain a sparse fine prediction

$$\hat{x}_{\mathrm{f}}^{T+1} = \mathcal{D}_{\mathrm{f}}(\widetilde{\mathbf{Z}}_{\mathrm{f}}^T) \in \mathbb{R}^{H \times W \times C}.$$

Finally, we fuse the full-resolution coarse prediction and the sparse fine prediction on the original grid using a lightweight CNN,

$$\hat{x}^{T+1} = \mathcal{R}\big(\hat{x}_{\mathrm{c}}^{T+1}, \hat{x}_{\mathrm{f}}^{T+1}\big),$$

where $\mathcal{R}$ concatenates the two predictions and produces the final next-step solution. This fusion step preserves global consistency from coarse prediction while injecting fine-scale corrections in refined regions.

### 3.3. Multi-Scale Position Encoding

Handling tokens with heterogeneous patch sizes requires a positional encoding scheme that is consistent across scales and robust to non-uniform patch partitioning. To this end, we construct a resolution-aware positional encoding that explicitly captures both spatial location and scale level, and apply it only once before tokens are fed into the Transformer.

We normalize the spatial domain to a unit square $[0, 1] \times [0, 1]$. Let $N_y = H/p$, $N_x = W/p$ and $\Delta_{c,x} = \frac{1}{N_x}$, $\Delta_{c,y} = \frac{1}{N_y}$. At the coarse scale, the domain is partitioned into an $N_y \times N_x$ patch grid. We represent each coarse patch by the center of its cell,

$$x_i = \left(j + \tfrac{1}{2}\right)\Delta_{c,x}, \quad y_i = \left(i + \tfrac{1}{2}\right)\Delta_{c,y}$$

where $(i, j)$ denotes the patch index on the coarse grid. At the fine scale, we use patches of size $(p/2) \times (p/2)$, which corresponds to a $2N_y \times 2N_x$ grid of cell centers with step sizes

$$\Delta_{f,x} = \frac{1}{2N_x}, \qquad \Delta_{f,y} = \frac{1}{2N_y}.$$

In this way, every token (coarse or fine) is assigned a continuous coordinate

$$\mathbf{p}_i = (x_i, y_i) \in [0, 1]^2.$$

In addition to spatial location, we encode the scale level to disambiguate tokens that may share similar coordinates but differ in patch size. We assign a scale indicator

$$d_i = \begin{cases} 0, & \text{coarse token,} \\ 1, & \text{fine token,} \end{cases}$$

and treat it as an explicit component of the positional representation.

For the unified multi-scale token sequence at time step $t$,

$$\mathbf{Z}^t \in \mathbb{R}^{(N_c + 3m) \times d}, \qquad N_c = N_x N_y, \ \ m = \lfloor k N_c \rfloor,$$

we map each token's positional tuple $(x_i, y_i, d_i)$ through a lightweight MLP $\phi(\cdot)$ to produce FiLM (Perez et al., 2018) parameters

$$(\boldsymbol{\gamma}_i, \boldsymbol{\beta}_i) = \phi(x_i, y_i, d_i), \qquad \boldsymbol{\gamma}_i, \boldsymbol{\beta}_i \in \mathbb{R}^d.$$

Instead of additive positional embeddings, we modulate token features via a feature-wise affine transformation,

$$\mathbf{Z}_i^{t,0} = \boldsymbol{\gamma}_i \odot \mathbf{Z}_i^t + \boldsymbol{\beta}_i,$$

which is applied once before entering the Transformer and kept fixed across layers. This resolution-aware FiLM conditioning provides a coherent notion of spatial locality and scale, enabling consistent interactions among tokens at different patch sizes under adaptive refinement.

## 4. Experiments

We evaluate the proposed MeshTok model through a comprehensive set of experiments designed to assess its accuracy, generalization, scalability, and computational efficiency. Specifically, our evaluation includes: (1) comparisons with uniform-patch Transformers and other baselines; (2) transfer performance on downstream tasks; (3) an analysis of scaling behavior with respect to model size and token budget; (4) quantitative measurements of inference-time computational cost; (5) an additional evaluation of model extension to 3D PDE data; and (6) ablation studies examining the impact of key hyperparameters. Unless otherwise stated, all relative errors in this section and the appendix are reported in percentage points, with the percent sign omitted.

**Pretraining Datasets**   During pretraining, we leverage a heterogeneous collection of PDE datasets drawn from PDEBench (Takamoto et al., 2022), PDENNEval (Wei et al., 2024), and The Well (Ohana et al., 2024). The mathematical formulations of the underlying PDEs are summarized in Appendix B.1. To ensure consistency across datasets with diverse spatial resolutions, domains, and variable dimensions, we adopt a unified preprocessing pipeline based on padding and masking. A detailed description of the preprocessing procedures is deferred to Appendix B.2.

**Training Setup**   Unless otherwise specified, all model architectures and hyperparameter configurations are provided in Appendix C. We train all models using the AdamW optimizer (Loshchilov & Hutter, 2017) with a learning rate of $1 \times 10^{-4}$ and a batch size of 8. Pretraining is conducted on a single NVIDIA A800 GPU for 20 epochs, where each epoch consists of 4,000 optimization steps (i.e., 80,000 steps in

*Table 1.* We conduct pretraining experiments by training all models from scratch on multiple datasets under a unified *big* configuration, resulting in comparable model sizes. Model performance is assessed using relative $\ell_2$ error, with lower values corresponding to better predictive accuracy (The symbol "-" in the table indicates that the relative error exceeds $50\%$, in which case the result is no longer practically meaningful).

| Model | Params (M) | The Well | PDENNeval | PDEBench | | |
|---|---|---|---|---|---|---|
| | | gray-scott | allen-cahn | CNS (1.0,0.01) | CNS (0.1,0.01) | SWE |
| DeepONet (Lu et al., 2019) | 17.26 | 30.983 | – | 13.357 | 4.589 | 8.439 |
| FNO (Li et al., 2021) | 21.04 | 15.639 | 3.886 | 4.998 | 1.418 | 1.098 |
| ViT (Dosovitskiy, 2020) | 45.25 | 6.471 | 2.809 | 2.808 | 0.637 | 0.647 |
| MPP (Masliaev et al., 2025) | 34.58 | 5.086 | 2.917 | 5.307 | 0.987 | 1.336 |
| DPOT (Hao et al., 2024) | 26.47 | 5.199 | 2.481 | 3.844 | 0.854 | 0.790 |
| MoE-POT (Wang et al., 2025a) | 59.15 | 5.971 | 2.428 | 4.801 | 0.863 | 0.757 |
| BCAT (Liu et al., 2025) | 29.19 | 2.345 | 1.310 | 1.339 | 0.254 | 0.443 |
| Ours | 31.43 | **2.095** | **1.056** | **1.187** | **0.249** | **0.276** |

total). For downstream tasks, we fine-tune the pretrained models for 5 epochs under the same training schedule (i.e., 20,000 steps in total). Detailed descriptions of the training protocol, evaluation procedure, and implementation settings are deferred to Appendix D.

**Baselines** As foundation-model baselines, we compare our approach with several representative architectures, including ViT (Dosovitskiy, 2020), MPP (Masliaev et al., 2025), DPOT (Hao et al., 2024), BCAT (Liu et al., 2025), and MoE-POT (Wang et al., 2025a). We additionally include established PDE learning models such as FNO (Li et al., 2021) and DeepONet (Lu et al., 2019) as comparative baselines. All models are trained and evaluated using comparable data splits and closely matched optimization settings to ensure a fair and meaningful comparison.

In addition, we investigate how different architectural choices affect refinement behavior, with detailed results reported in Appendix E.

### 4.1. Multi-Dataset Training Results

We evaluate our method against several representative PDE foundation baselines on multiple public benchmarks in the pretraining setting. As shown in Table 1, our model achieves consistently low relative errors across The Well, PDEN-NEval, and PDEBench under comparable training conditions, and is competitive with prior foundation models on all evaluated tasks.

Across these benchmarks, conventional neural operator baselines (e.g., DeepONet and FNO) attain reasonable accuracy on some settings, but their performance varies noticeably across PDE families and parameter regimes. In particular, on more challenging fluid dynamics benchmarks in PDEBench (CNS and SWE), their relative errors are substantially higher than those of recent foundation models, and some results are not reported (marked as "–"). In contrast,

our method yields the lowest errors among the compared approaches on both CNS and SWE in PDEBench, and also improves over previous methods on Gray–Scott in The Well and Allen–Cahn in PDENNEval. Overall, these results suggest improved transferability across different equations and datasets, while maintaining stable accuracy across heterogeneous data distributions.

We hypothesize that these improvements are supported by three design components. First, the coarse-to-fine encoder–decoder provides multi-resolution representations that jointly capture smooth global structures and locally complex regions, leading to more diverse and comprehensive spatial features. Second, the spatiotemporal block-causal self-attention introduces a structured inductive bias for modeling long-horizon temporal evolution while preserving full spatial interactions within each time step. Third, the adaptive patch refinement mechanism allocates additional resolution to regions with higher local complexity, which can improve fine-scale accuracy without uniformly increasing computation.

### 4.2. Downstream Fine-Tuning

We evaluate whether pretraining on source equations helps MeshTok adapt to *unseen PDE families* via downstream fine-tuning (Zhou et al., 2024). To ensure a strict transfer setting, we select four downstream equations that are not included during pre-training: Reaction–Diffusion from PDEBENCH, Shear flow from THE WELL, and Burgers and Black–Scholes–Barenblatt from PDENNEVAL. These tasks therefore test cross-equation generalization rather than in-domain memorization.

We compare MeshTok with the BCAT baseline under two regimes: **Scratch** and **Pretrained → finetune**. For each downstream PDE, both models use the same downstream data, optimization hyperparameters, and fine-tuning budget of 5 epochs. Figure 3 summarizes the fine-tuning curves,

and Table 2 reports the final errors at epoch 5.

Overall, pretraining improves both BCAT and MeshTok on three out of four downstream PDE families. For Mesh-Tok, the epoch-5 error decreases on Shear flow ($3.817 \rightarrow 3.334$), Burgers ($0.324 \rightarrow 0.252$), and Reaction–Diffusion ($11.096 \rightarrow 8.106$). A similar trend holds for BCAT on the same three tasks. On Black–Scholes–Barenblatt, pretraining does not improve the final error for either model, likely due to the larger distribution shift from the physical systems dominating the pretraining corpus. Across fine-tuned models, MeshTok achieves the lowest error on three out of four downstream PDE families, while BCAT trained from scratch remains best on Black–Scholes–Barenblatt.

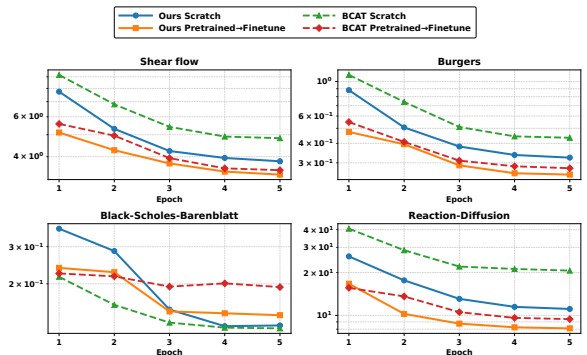

*Figure 3.* Benefit of pretraining across four PDE families. Each subplot compares training from scratch against fine-tuning from a pretrained model for BCAT and MeshTok. Lower relative $\ell_2$ error is better.

These results suggest that pretraining can provide a useful initialization that transfers beyond the source equations and improves sample efficiency in several downstream settings. They also show that MeshTok retains a favorable advantage over the BCAT baseline after fine-tuning on most unseen PDE families.

### 4.3. Refinement Gains and Scaling Behavior

We study how spatial refinement interacts with model scaling, and whether refinement benefits persist across different capacity regimes. To this end, we evaluate three model scales—OURS-SMALL, OURS-BIG, and OURS-LARGE—which share the same architecture and training setup but differ only in network width and depth. The specific model configuration can be found in Table 8. For each scale, we compare three refinement settings: **No refinement** (uniform coarse partition), **Ours (activity-based)** refinement (our default policy that refines a fixed fraction of patches based on gradient/energy activity), and **Full refinement** (uniformly refining all patches to the finest resolution). We report averaged relative $\ell_2$ errors for both **1-step** prediction and **10-step** autoregressive rollout. Quantitative results are

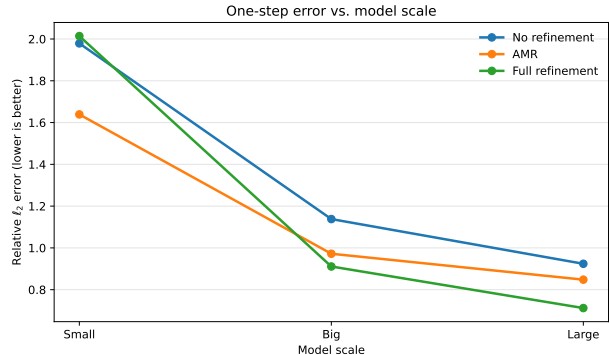

*Figure 4.* One-step prediction error versus model scale under different refinement settings.

shown in Table 3, and the corresponding 1-step scaling trend is visualized in Figure 4.

*Table 3.* Averaged relative $\ell_2$ error (lower is better) across all benchmark datasets under different refinement settings and model scales.

| Refinement | Horizon | Model scale | | |
|---|---|---|---|---|
| | | SMALL | BIG | LARGE |
| No refinement | 1-step | 1.979 | 1.138 | 0.924 |
| | 10-step | 4.894 | 2.607 | 2.125 |
| Ours | 1-step | 1.639 | 0.972 | 0.837 |
| | 10-step | 4.027 | 2.261 | 1.966 |
| Full refinement | 1-step | 2.014 | 0.911 | 0.716 |
| | 10-step | 5.287 | 2.221 | 1.696 |

Table 3 shows a consistent scaling behavior: increasing model capacity reduces both 1-step and 10-step errors across all refinement settings. For example, under AMR refinement, the 1-step error decreases from 1.639 (SMALL) to 0.972 (BIG) and further to 0.837 (LARGE), and the 10-step error decreases from 4.027 to 2.261 and 1.966, respectively. This indicates that the proposed model benefits from additional capacity in both short-horizon prediction and long-horizon rollout.

At each model scale, *Ours* consistently improves over *No refinement* for both horizons, indicating that refining a subset of patches based on physical activity is beneficial across capacity regimes. For BIG and LARGE, *Full refinement* yields the lowest 1-step errors (0.911 and 0.716), while *Ours* remains close (0.972 and 0.837) and also improves 10-step rollout relative to *No refinement* (2.261 vs. 2.607 for BIG, and 1.966 vs. 2.125 for LARGE). These results suggest that partial refinement can capture a substantial portion of the refinement benefit while preserving efficiency.

*Table 2.* Fine-tuning results at epoch 5 across four PDE families. We compare the BCAT baseline and MeshTok under two settings: training from scratch and fine-tuning from a pre-trained initialization. Lower relative $\ell_2$ error indicates better prediction accuracy.

| Model | Setting | Shear flow | Burgers | Black–Scholes–Barenblatt | React–Diff |
|---|---|---|---|---|---|
| BCAT | Scratch | 4.827 | 0.435 | **0.123** | 20.626 |
| | Pretrained → finetune | 3.487 | 0.277 | 0.193 | 9.414 |
| Ours | Scratch | 3.817 | 0.324 | 0.127 | 11.096 |
| | Pretrained → finetune | **3.334** | **0.252** | 0.142 | **8.106** |

An important observation is that OURS-SMALL does not always benefit from uniform *Full refinement*. In particular, *Full refinement* increases the 1-step error from 1.979 to 2.014 and the 10-step error from 4.894 to 5.287, suggesting that the smallest Transformer does not have sufficient capacity to effectively model and exploit a globally high-resolution token sequence. By contrast, *Ours* achieves substantially lower errors at the same scale (1-step: 1.639; 10-step: 4.027), outperforming *Full refinement* on both horizons. We attribute this behavior to our coarse–fine dual encoder–decoder, which provides a structured multi-resolution representation that explicitly assists the Transformer backbone as the primary modeling component. Specifically, it preserves a compact coarse representation for global context, while selectively injecting fine tokens only in high-activity regions, so that the Transformer can focus its limited capacity on the most informative details without being overwhelmed by uniformly dense tokens.

Overall, across all three scales, activity-guided refinement provides consistent improvements over coarse tokenization, and its benefits remain compatible with model scaling. A three-seed robustness study in Appendix F.1 further confirms that AMR consistently improves over no refinement in this scaling comparison, with small standard deviations across runs.

### 4.4. Trade-Off between Accuracy and Efficiency

We further evaluate the trade-off between accuracy and efficiency of refinement under a fixed model capacity, and use the OURS-BIG configuration as a representative example. We compare three inference settings in a consistent order throughout this section: **No refinement** (coarse tokens only), **Ours** (AMR with activity-based partial refinement), and **Full refinement** (uniform refinement everywhere).

Figure 5 summarizes the Pareto-style trade-off between per-step runtime and one-step prediction error under the OURS-BIG model. The plot includes a baseline line connecting *No refinement* and *Full refinement*, representing the reference trade-off achieved by uniformly changing the refinement level. The gray region highlights configurations that dominate this baseline (simultaneously lower error and lower runtime), and the arrow indicates the direction of

*Table 4.* Compute-matched comparison under the same data setting and training protocol. Width or depth is adjusted so that the MACs are close to MeshTok. Lower relative error is better.

| Model | $N$/step | Arch. $(L, d, f; p)$ | MACs | Params (M) | Rel. ↓ |
|---|---|---|---|---|---|
| Coarse, width increased | 64 | $(8, 640, 1800; 8)$ | $2.650\times10^9$ | 47.19 | 1.0294 |
| Coarse, depth increased | 64 | $(14, 512, 1280; 8)$ | $2.760\times10^9$ | 47.33 | 1.0652 |
| MeshTok, activity-based | 112 | $(8, 512, 1280; 8)$ | $2.804\times10^9$ | 31.43 | **0.9723** |
| Full refine, width reduced | 256 | $(8, 320, 800; 16)$ | $2.747\times10^9$ | 10.38 | 1.0777 |
| Full refine, depth reduced | 256 | $(4, 512, 1280; 16)$ | $3.355\times10^9$ | 13.67 | 1.0632 |

improvement (faster and more accurate). Our method lies in this favorable region by achieving lower error than *No refinement* while requiring substantially less time than *Full refinement*, illustrating an improved accuracy–efficiency balance enabled by activity-guided partial refinement.

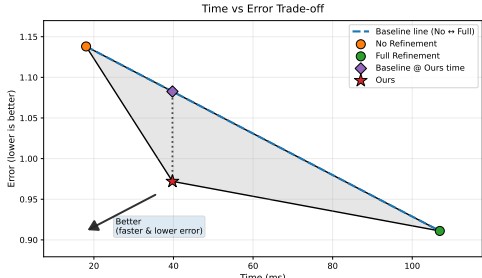

*Figure 5.* **Time vs. error trade-off under OURS-BIG.** Each point corresponds to one refinement setting: *No refinement*, *Ours* (activity-based AMR), and *Full refinement*. The baseline line interpolates the reference trade-off between *No refinement* and *Full refinement*. The gray region indicates configurations that are strictly better than the baseline, and the arrow marks the direction of improvement (faster and lower error).

To further check whether the improvement is simply due to using more computation, we conduct a compute-matched comparison by adjusting the width or depth of uniform-token baselines so that their MACs are close to MeshTok. All variants use the same data setting and training protocol, and differ only in how the comparable compute budget is allocated across tokens and backbone capacity.

Table 4 shows that MeshTok achieves the lowest relative error under matched-MAC accounting. The comparison includes stronger coarse-token baselines with increased width or depth, as well as full-refinement baselines whose width or

depth is reduced to keep the MACs comparable. Therefore, the gain is not solely explained by larger token count or backbone compute under this accounting; rather, adaptive tokenization provides a more effective allocation of computation to locally active regions. This MAC-based comparison is complementary to the wall-clock runtime results, since actual latency also depends on hardware utilization and implementation details.

To complement the trade-off view, we further report the averaged errors and the corresponding runtime measurements under OURS-BIG. The error results are shown in Table 3 and the runtime results are shown in Table 6, both ordered as *No refinement → Ours → Full refinement*. Consistent with the trade-off curve, partial refinement reduces error compared to the coarse baseline, while avoiding the higher cost of uniformly refining all patches. Additional analyses and trade-off results under other model scales and refinement ratios are provided in Appendix A.2.

### 4.5. Additional 3D Experiment

To further examine whether the refinement mechanism remains useful beyond 2D fields, we conduct an additional 3D experiment on The Well MHD dataset. This experiment uses 100 trajectories with four physical variables, namely density and the three velocity components. Because this 3D dataset is smaller than our main training sets, all models are trained for 5 epochs with 1,000 iterations per epoch.

In 3D, refining one coarse token produces eight fine tokens. We therefore use a refinement ratio of $k = 0.375$, giving the AMR model a sequence length of 232, compared with 512 for full refinement. Since self-attention scales quadratically with sequence length, this corresponds to approximately $(232/512)^2 \approx 20.5\%$ of the dominant attention cost of full refinement.

*Table 5.* Additional 3D MHD results. Lower relative error and runtime are better.

| Metric | No refinement | AMR | Full refinement |
|---|---|---|---|
| Relative error ↓ | 39.647 | 35.851 | 29.626 |
| Runtime (ms) ↓ | 62.17 | 149.91 | 451.67 |

AMR improves over no refinement in this 3D setting while remaining substantially cheaper than full refinement. Full refinement achieves the lowest error, but AMR is about $3.0\times$ faster in wall-clock runtime and uses less than half the sequence length. These results provide preliminary 3D evidence for the same accuracy–efficiency trend observed in the main 2D experiments.

### 4.6. More Ablation Studies

Due to space constraints, we defer additional ablations to the appendix, including studies on refinement ratios, input resolutions, positional encoding designs, and other training details. Please refer to Appendix F for comprehensive results and discussions.

### 4.7. Limitations

The current implementation of MeshTok is designed for structured-grid PDE data, where coarse and fine tokens can be defined by regular spatial subdivisions. This design makes the refinement budget, token count, and positional encoding straightforward to control, but it also ties the method to Cartesian-style patch hierarchies. Extending the same adaptive-tokenization principle to unstructured meshes or irregular geometries is therefore nontrivial, since the patch hierarchy, neighborhood relations, and positional encodings are no longer given by a regular grid. Such settings would likely require graph- or mesh-based backbones, together with refinement rules and geometry-aware encodings designed for irregular connectivity.

This structured two-level design also leaves room for improving the refinement strategy itself. While our activity-based refinement remains effective across model scales, we observe a reduced marginal gain from adaptivity as model size increases, likely because larger models have stronger capacity to capture fine-scale structures even under coarser tokenization. Future work could explore scale-aware refinement, multi-level adaptive token hierarchies, or jointly optimizing the refinement policy with model capacity.

## 5. Conclusion

We presented **MeshTok**, a multi-scale tokenization framework that integrates adaptive mesh refinement (AMR) into Transformer-based PDE foundation models. By replacing uniform patchification with AMR-aware tokenization, MeshTok yields heterogeneous tokens that retain coarse global context while selectively resolving fine-scale structures in physically salient regions, yet remains fully compatible with standard Transformer architectures. It approaches full-refinement accuracy at substantially lower inference cost, exhibiting favorable scaling across model sizes and a practical runtime "cost attenuation" effect. These results highlight adaptive resolution as a principled mechanism for allocating computation in data-driven PDE modeling. Future work includes extending tokenization to unstructured and multi-physics settings, and integrating long-horizon stabilization objectives.

## Impact Statement

**MeshTok** proposes a multi-scale tokenization framework for Transformer-based PDE modeling, aiming to improve the trade-off between accuracy and efficiency of neural simulators for physical systems. By allocating computation adaptively to spatial regions with higher solution complexity, MeshTok can enable faster inference under fixed compute budgets, potentially benefiting scientific workflows that rely on repeated PDE solves, such as climate and weather modeling, fluid dynamics, energy systems, and materials simulation. Improved efficiency may also reduce the energy cost of training and deploying PDE models at scale, supporting more accessible and sustainable research and engineering applications.

At the same time, faster and more scalable PDE surrogates may lower the barrier to high-fidelity simulation and could be used in high-stakes engineering optimization. In addition, like other data-driven simulators, MeshTok inherits limitations from training data and evaluation benchmarks: models may generalize poorly outside the distribution of observed PDE regimes, potentially producing overconfident predictions in safety-critical settings. These risks motivate careful reporting of training domains, uncertainty estimation where appropriate, and validation against trusted numerical solvers before deployment.

We view MeshTok as a general computational principle—adaptive, multi-scale tokenization—rather than a domain-specific system. Future work may further improve responsible use by integrating reliability measures (e.g., uncertainty-aware refinement, out-of-distribution detection, and activity-based consistency checks), and by documenting intended use cases and failure modes. Overall, we expect MeshTok to have a positive impact by enabling more efficient and scalable PDE modeling, while emphasizing that real-world adoption should follow established verification and safety practices in scientific computing.

## Acknowledgements

This work was supported in part by the Fundamental and Interdisciplinary Disciplines Breakthrough Plan of the Ministry of Education of China under Grant JYB2025XDXM113, in part by the Natural Science Foundation of Jiangsu Province under Grant BK20240462, and in part by the National Natural Science Foundation of China (NSFC) under Grant 12425113. The authors also acknowledge support from the Key Laboratory of the Ministry of Education for Mathematical Foundations and Applications of Digital Technology at the University of Science and Technology of China.

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

# A. Theoretical Analysis

## A.1. Theoretical Analysis of AMR

A substantial body of prior work has demonstrated that modern neural architectures—including convolutional encoder–decoder surrogates and neural-operator/Transformer-based models—can approximate PDE solution operators with high accuracy when supplied with sufficiently informative representations of the input fields and geometries. Representative examples include convolutional surrogate models for parametric PDEs and uncertainty quantification (Zhu & Zabaras, 2018), universal approximation results for operator learning (Lu et al., 2019; Gonon & Jacquier, 2025; Chen & Chen, 1995), neural-operator architectures such as Fourier Neural Operators (Li et al., 2021; Kovachki et al., 2023), and attention/Transformer operator learners (Cao, 2021; Li et al., 2022a; Wang et al., 2025b; Ovadia et al., 2024).

Motivated by these results, our theoretical analysis adopts the following working premise: *the backbone model family (encoder–Transformer–decoder) is sufficiently expressive on the range of representations considered*, so that the dominant limitations arise not from inadequate depth/width, but from the *representation map* that converts raw PDE states into tokens. Concretely, once a partition $P$ (uniform or adaptive) is fixed, the learnable predictor is restricted to the realizable class $\mathcal{F}_P = \{D(T(\Phi_P(x)))\}$, where all dependence on the input $x$ is mediated by the tokenization $\Phi_P$.

These results should be viewed as a conditional representation analysis of the tokenization map, rather than as convergence guarantees for the full learning algorithm. Theorem A.1 identifies a token-collision lower bound that arises when $\Phi_P$ discards task-relevant information, independently of Transformer capacity. Theorem A.3 shows that, under an explicit alignment map and encoder/decoder compatibility assumptions, refinement can enlarge the realizable function class and cannot increase the best uniform approximation error. Theorem A.4 further relates AMR to the budgeted best approximation error under a near-optimal refinement assumption. Accordingly, our proofs focus on the representational consequences of $\Phi_P$ and its induced partition, rather than on universal approximation or training convergence of the Transformer backbone.

**Theorem A.1** (Token-collision lower bound under encoder–Transformer–decoder architectures)**.** *Let $\mathcal{X} \subset \mathbb{R}^{d_x}$ be a compact domain and let $g : \mathcal{X} \to \mathbb{R}^{d_y}$ be a target function. For a given patch partition $P$, denote by $\Phi_P : \mathcal{X} \to \mathbb{R}^{N(P) \times d}$ the associated deterministic tokenization map. Let $\mathcal{T}_{N,d}$ be a family of Transformer encoders operating on $N$ tokens of width $d$, and let $\mathcal{D}_{N,d}$ be a family of decoder heads mapping encoder outputs to $\mathbb{R}^{d_y}$.*

*Define the realizable function class*

$$\mathcal{F}_P := \big\{ f(x) = D(T(\Phi_P(x))) \ : \ T \in \mathcal{T}_{N(P),d}, \ D \in \mathcal{D}_{N(P),d} \big\},$$

*and the best uniform approximation error*

$$\mathcal{E}(P; g) := \inf_{f \in \mathcal{F}_P} \sup_{x \in \mathcal{X}} \|f(x) - g(x)\|.$$

*Suppose there exist two distinct points $x_0, x_1 \in \mathcal{X}$ such that*

$$\Phi_P(x_0) = \Phi_P(x_1) \quad \textit{but} \quad g(x_0) \neq g(x_1).$$

*Then the following lower bound holds:*

$$\mathcal{E}(P; g) \ \geq \ \frac{1}{2} \|g(x_0) - g(x_1)\|.$$

*Proof.* We begin by observing that the tokenization map $\Phi_P$ induces an equivalence relation on $\mathcal{X}$: two inputs $x, x' \in \mathcal{X}$ are equivalent if $\Phi_P(x) = \Phi_P(x')$. All functions $f \in \mathcal{F}_P$ are constant on each such equivalence class, since they depend on $x$ only through $\Phi_P(x)$.

Fix an arbitrary function $f \in \mathcal{F}_P$. By definition, there exist $T \in \mathcal{T}_{N(P),d}$ and $D \in \mathcal{D}_{N(P),d}$ such that

$$f(x) = D(T(\Phi_P(x))).$$

Because $\Phi_P(x_0) = \Phi_P(x_1)$, we necessarily have

$$T(\Phi_P(x_0)) = T(\Phi_P(x_1)),$$

and therefore
$$f(x_0) = f(x_1) =: c \in \mathbb{R}^{d_y}.$$

Since $g(x_0) \neq g(x_1)$, any constant value $c$ must incur a nonzero error on at least one of the two points. By the triangle inequality,
$$\|g(x_0) - g(x_1)\| \leq \|g(x_0) - c\| + \|c - g(x_1)\| \leq 2 \max\{\|g(x_0) - c\|, \ \|g(x_1) - c\|\}.$$

Consequently,
$$\max\{\|f(x_0) - g(x_0)\|, \ \|f(x_1) - g(x_1)\|\} \ \geq \ \frac{1}{2} \|g(x_0) - g(x_1)\|.$$

Since the supremum over $\mathcal{X}$ is no smaller than the maximum over $\{x_0, x_1\}$, we obtain
$$\sup_{x \in \mathcal{X}} \|f(x) - g(x)\| \ \geq \ \frac{1}{2} \|g(x_0) - g(x_1)\|.$$

Finally, taking the infimum over all $f \in \mathcal{F}_P$ yields the claimed lower bound on $\mathcal{E}(P; g)$. $\qquad\square$

*Remark* A.2 (Patch-induced information bottleneck). The above theorem formalizes a fundamental expressivity limitation imposed by the patch partition $P$ (Shannon, 1949). In practice, coarse patch tokenization schemes typically retain only low-dimensional summaries of each patch, such as linear projections, averages, or pooled statistics. As a result, distinct inputs that differ in fine-scale details may collapse to the same token representation. Our coarse–fine dual encoder–decoder is designed to mitigate this issue by extracting richer multi-resolution information during tokenization, so that less task-relevant content is discarded when forming patch-level representations.

If the target function $g$ depends on information discarded by $\Phi_P$, then no choice of Transformer depth, width, or nonlinearity can compensate for this loss: the model class $\mathcal{F}_P$ is intrinsically incapable of separating inputs within the same token equivalence class. This yields a hard lower bound on the achievable approximation error, determined solely by the patch partition rather than the capacity of the Transformer itself.

**Theorem A.3** (Refinement implies realizable-class inclusion via a nonlinear alignment map). *Let $P_1 \preceq P_2$ be two patch partitions, where $P_2$ is a refinement of $P_1$. Let $\Phi_{P_i} : \mathcal{X} \to \mathbb{R}^{N(P_i) \times d}$ be the corresponding tokenization maps. Assume there exists a (possibly nonlinear) alignment map*
$$R : \mathrm{Range}(\Phi_{P_2}) \to \mathbb{R}^{N(P_1) \times d}$$

*such that*
$$\Phi_{P_1}(x) = R(\Phi_{P_2}(x)), \qquad \forall x \in \mathcal{X}. \tag{1}$$

*Let $\mathcal{T}_{N,d}$ be a family of Transformer encoders operating on $N$ tokens of width $d$, and let $\mathcal{D}_{N,d}$ be a family of decoder heads mapping encoder outputs to $\mathbb{R}^{d_y}$. Define the realizable classes*
$$\mathcal{F}_{P_i} := \Big\{ f(x) = D(T(\Phi_{P_i}(x))) \ : \ T \in \mathcal{T}_{N(P_i),d}, \ D \in \mathcal{D}_{N(P_i),d} \Big\}.$$

*Because the corresponding function family obtained from the refined division is sufficiently rich compared to the coarse division, assume the following closure/compatibility properties:*

*(A1′) Transformer closure under precomposition with $R$. For every $T_1 \in \mathcal{T}_{N(P_1),d}$, there exists $T_2 \in \mathcal{T}_{N(P_2),d}$ such that*
$$T_2(z) = T_1(R(z)), \qquad \forall z \in \mathrm{Range}(\Phi_{P_2}).$$

*(A2′) Decoder compatibility on the realized encoder range. For every $D_1 \in \mathcal{D}_{N(P_1),d}$, there exists $D_2 \in \mathcal{D}_{N(P_2),d}$ such that*
$$D_2(y) = D_1(y), \qquad \forall y \in \mathrm{Range}\big(T_1 \circ \Phi_{P_1}\big).$$

*Then the realizable classes satisfy the inclusion*
$$\mathcal{F}_{P_1} \subseteq \mathcal{F}_{P_2}.$$

*Consequently, for any target function $g : \mathcal{X} \to \mathbb{R}^{d_y}$, the optimal uniform approximation error is monotone under refinement:*
$$\mathcal{E}(P_2; g) \leq \mathcal{E}(P_1; g), \quad \text{where} \quad \mathcal{E}(P; g) := \inf_{f \in \mathcal{F}_P} \sup_{x \in \mathcal{X}} \|f(x) - g(x)\|.$$

*Proof.* We prove $\mathcal{F}_{P_1} \subseteq \mathcal{F}_{P_2}$ by constructing, for an arbitrary $f \in \mathcal{F}_{P_1}$, an explicit witness representation of $f$ inside $\mathcal{F}_{P_2}$.

First, fix an arbitrary function in the coarse realizable class. Let $f \in \mathcal{F}_{P_1}$. By definition, there exist $T_1 \in \mathcal{T}_{N(P_1),d}$ and $D_1 \in \mathcal{D}_{N(P_1),d}$ such that

$$f(x) = D_1(T_1(\Phi_{P_1}(x))), \qquad \forall x \in \mathcal{X}. \tag{2}$$

Then, by the alignment identity (1), we can express the coarse token stream as a deterministic transformation of the refined token stream:

$$\Phi_{P_1}(x) = R(\Phi_{P_2}(x)).$$

Substituting this into (2) yields

$$f(x) = D_1(T_1(R(\Phi_{P_2}(x)))). \tag{3}$$

At this point, the expression is still not in the canonical form $D_2(T_2(\Phi_{P_2}(x)))$, because the composition $T_1 \circ R$ may not itself lie in $\mathcal{T}_{N(P_2),d}$ a priori.

Apply assumption (A1') to $T_1$. It guarantees the existence of some $T_2 \in \mathcal{T}_{N(P_2),d}$ such that

$$T_2(z) = T_1(R(z)), \qquad \forall z \in \operatorname{Range}(\Phi_{P_2}).$$

In particular, for every $x \in \mathcal{X}$, we have $\Phi_{P_2}(x) \in \operatorname{Range}(\Phi_{P_2})$, hence

$$T_2(\Phi_{P_2}(x)) = T_1(R(\Phi_{P_2}(x))) = T_1(\Phi_{P_1}(x)). \tag{4}$$

The equality (4) shows that the encoder outputs produced by the refined pipeline $T_2 \circ \Phi_{P_2}$ coincide (pointwise on $\mathcal{X}$) with those produced by the coarse pipeline $T_1 \circ \Phi_{P_1}$. Now invoke assumption (A2') for the given $D_1$ (and the already fixed $T_1$): there exists $D_2 \in \mathcal{D}_{N(P_2),d}$ such that

$$D_2(y) = D_1(y), \qquad \forall y \in \operatorname{Range}(T_1 \circ \Phi_{P_1}).$$

Since $T_1(\Phi_{P_1}(x)) \in \operatorname{Range}(T_1 \circ \Phi_{P_1})$ for all $x \in \mathcal{X}$, we obtain

$$D_2(T_1(\Phi_{P_1}(x))) = D_1(T_1(\Phi_{P_1}(x))), \qquad \forall x \in \mathcal{X}. \tag{5}$$

And combining (2), (4), and (5), for every $x \in \mathcal{X}$,

$$f(x) = D_1(T_1(\Phi_{P_1}(x))) = D_2(T_1(\Phi_{P_1}(x))) = D_2(T_2(\Phi_{P_2}(x))).$$

Thus $f$ admits a representation of the form $f(x) = D_2(T_2(\Phi_{P_2}(x)))$ with $T_2 \in \mathcal{T}_{N(P_2),d}$ and $D_2 \in \mathcal{D}_{N(P_2),d}$, i.e. $f \in \mathcal{F}_{P_2}$. Since $f \in \mathcal{F}_{P_1}$ was arbitrary, we have shown $\mathcal{F}_{P_1} \subseteq \mathcal{F}_{P_2}$.

Finally, because $\mathcal{F}_{P_1} \subseteq \mathcal{F}_{P_2}$, the infimum of the same functional over the larger set cannot be larger:

$$\inf_{f \in \mathcal{F}_{P_2}} \sup_{x \in \mathcal{X}} \|f(x) - g(x)\| \;\leq\; \inf_{f \in \mathcal{F}_{P_1}} \sup_{x \in \mathcal{X}} \|f(x) - g(x)\|.$$

This is exactly $\mathcal{E}(P_2; g) \leq \mathcal{E}(P_1; g)$. $\qquad\square$

**Theorem A.4** (Rationality of AMR partitions under a shared Transformer backbone). *Let $P_1$ be a uniform patch partition and let $P_2$ be an adaptive mesh refinement (AMR) partition constructed by an error indicator. Assume refinement is applied only to a strict subset of patches and satisfies the budget constraint*

$$N(P_2) \;\leq\; C\,N(P_1),$$

*for some constant $C > 1$ depending only on the refinement rule.*

*Assume that the realizable-class inclusion result of Theorem A.3 holds for the pair $(P_1, P_2)$, i.e., there exists a (possibly nonlinear) alignment map $R$ and the encoder/decoder families satisfy assumptions (A1')–(A2'). Assume further that the target function $g : \mathcal{X} \to \mathbb{R}^{d_y}$ exhibits* nonuniform local complexity, *meaning that its best achievable approximation error*

*under a patch partition concentrates on a strict subset of spatial regions. Assume the AMR indicator is equivalent to the local complexity density of g and therefore refines precisely those high-complexity regions.*

*Define the budgeted best approximation error at token budget $N(P_2)$ as*

$$\mathcal{E}^{\star}(N(P_2); g) := \inf_{\tilde{P}: N(\tilde{P}) \leq N(P_2)} \mathcal{E}(\tilde{P}; g).$$

*Assume the AMR rule is* budget-near-optimal, *i.e., there exists $C_{\mathrm{qo}} \geq 1$ such that*

$$\mathcal{E}(P_2; g) \leq C_{\mathrm{qo}} \mathcal{E}^{\star}(N(P_2); g).$$

*Then the following statements hold:*

1. $\mathcal{F}_{P_1} \subsetneq \mathcal{F}_{P_2}$;

2. *If $C_{\mathrm{qo}} = 1$, then*
$$\mathcal{E}(P_2; g) = \mathcal{E}^{\star}(N(P_2); g) \leq \mathcal{E}(P_1; g);$$

3. *More generally, for arbitrary $C_{\mathrm{qo}} \geq 1$,*
$$\mathcal{E}(P_2; g) \leq C_{\mathrm{qo}} \mathcal{E}^{\star}(N(P_2); g) \leq C_{\mathrm{qo}} \mathcal{E}(P_1; g).$$

*Proof.* We decompose the argument into four conceptual steps.

Since refinement never reduces the token count, we have $N(P_1) \leq N(P_2)$. Therefore, the uniform partition $P_1$ is a feasible candidate in the definition of $\mathcal{E}^{\star}(N(P_2); g)$. By definition of the infimum,

$$\mathcal{E}^{\star}(N(P_2); g) = \inf_{\tilde{P}: N(\tilde{P}) \leq N(P_2)} \mathcal{E}(\tilde{P}; g) \leq \mathcal{E}(P_1; g).$$

By the budget-near-optimality assumption, the approximation error achieved by the AMR partition $P_2$ satisfies

$$\mathcal{E}(P_2; g) \leq C_{\mathrm{qo}} \mathcal{E}^{\star}(N(P_2); g).$$

If $C_{\mathrm{qo}} = 1$, then $P_2$ attains the optimal error among all partitions under the same token budget. Combining with Step 1 yields

$$\mathcal{E}(P_2; g) = \mathcal{E}^{\star}(N(P_2); g) \leq \mathcal{E}(P_1; g),$$

and the general case follows by transitivity of inequalities.

By Theorem A.3, we already have $\mathcal{F}_{P_1} \subseteq \mathcal{F}_{P_2}$. We now argue that the inclusion is strict. Because $g$ has nonuniform local complexity and the AMR indicator refines precisely the high-complexity regions, there exist $x_0, x_1 \in \mathcal{X}$ such that

$$\Phi_{P_1}(x_0) = \Phi_{P_1}(x_1), \qquad \Phi_{P_2}(x_0) \neq \Phi_{P_2}(x_1), \qquad g(x_0) \neq g(x_1).$$

Under $P_1$, the pair $(x_0, x_1)$ is indistinguishable at the token level, whereas under $P_2$ is separated by refinement.

By construction, any $f \in \mathcal{F}_{P_1}$ must satisfy $f(x_0) = f(x_1)$, and therefore incurs a nonzero worst-case error on $\{x_0, x_1\}$. In contrast, under the refined representation $\Phi_{P_2}$, the shared Transformer backbone and decoder family are assumed expressive enough to distinguish the separated tokens. Hence there exists $f_2 \in \mathcal{F}_{P_2}$ that matches $g$ on $\{x_0, x_1\}$ up to arbitrary accuracy. This proves $\mathcal{F}_{P_1} \subsetneq \mathcal{F}_{P_2}$.

A fully refined partition (uniformly refining all patches) would generally achieve a smaller or equal approximation error than $P_2$, as it enlarges the feasible set in the definition of $\mathcal{E}^{\star}$. However, such *Full refinement* typically requires a token count far exceeding $N(P_2)$ and therefore incurs prohibitive computational cost. The AMR partition $P_2$ can thus be interpreted as a budget-aware compromise: it improves expressivity over $P_1$ while avoiding the unnecessary complexity of global refinement. □

*Remark* A.5 (Interpretation and practical implication). The theorem establishes a three-way ordering: uniform partition $\prec$ AMR partition $\prec$ *Full refinement*. Under the stated assumptions, AMR strictly improves the approximation capability over uniform patching under the same Transformer backbone, yet remains computationally feasible. Its effectiveness depends critically on how well the refinement rule approximates the budgeted infimum $\mathcal{E}^\star(N; g)$; poor indicators lead to suboptimal partitions, whereas near-optimal indicators yield near-optimal performance. In practice, an *activity-based* rule can be interpreted as a tractable surrogate to this budgeted optimum: it assigns each patch $\mathcal{P}_i$ a score

$$s_i^{\mathrm{act}}(x) = \frac{1}{|\mathcal{P}_i|} \sum_{u \in \mathcal{P}_i} \psi(x(u)), \qquad \psi(x(u)) \in \{\|\nabla x(u)\|_2, \ \|\Delta x(u)\|_2^2, \ \text{or their combination}\},$$

and refines the top-$k$ patches with the largest $s_i^{\mathrm{act}}$. When $\psi(\cdot)$ correlates with the local approximation difficulty (i.e., higher error density or stronger small-scale variations), this heuristic allocation tends to place the limited refinement budget on regions that contribute most to $\mathcal{E}^\star(N; g)$ (Babuška & Rheinboldt, 1978; Ainsworth & Oden, 1997; Dörfler, 1996).

### A.2. The Efficiency of AMR

**Setting** We consider spatiotemporal inputs $x \in \mathbb{R}^{B \times T \times H \times W \times C}$. Each frame is tokenized by patch embedding, and tokens across $T$ frames are concatenated into a single sequence on which a shared Transformer backbone performs causal modeling for next-step prediction. We analyze the training/inference compute in FLOPs up to constant factors.

We focus exclusively on the dominant cost of causal self-attention and ignore all lower-order components (e.g., patch embedding, feed-forward networks, and decoders).

Let $x \in \mathbb{R}^{B \times T \times H \times W \times C}$ be the input. After patch tokenization, all tokens across $T$ frames are concatenated into a single sequence, on which a shared Transformer backbone applies causal self-attention for next-step prediction.

Let $p \times p$ denote the coarse patch size. The number of coarse tokens per frame is

$$N_c = \frac{H}{p} \cdot \frac{W}{p} = \frac{HW}{p^2}.$$

Using $(p/2) \times (p/2)$ patches over the full domain yields $N_f = 4N_c$ tokens per frame.

Under full fine tokenization, the sequence length is

$$S_f = TN_f = 4TN_c.$$

Under adaptive mesh refinement (AMR), a fraction $k \in [0, 1]$ of coarse patches is refined, where each refined coarse patch is replaced by four fine tokens. The resulting number of tokens per frame is

$$N_{\mathrm{amr}} = (1 - k)N_c + 4kN_c = (1 + 3k)N_c,$$

and the sequence length becomes

$$S_{\mathrm{amr}} = TN_{\mathrm{amr}} = (1 + 3k)TN_c.$$

For a sequence of length $S$ and embedding dimension $D$, the computational complexity of causal self-attention scales as

$$\mathcal{C}_{\mathrm{attn}}(S) = \mathcal{O}(S^2 D).$$

Therefore, the ratio between the AMR attention cost and the full fine-grid attention cost is

$$\frac{\mathcal{C}_{\mathrm{attn}}^{\mathrm{amr}}}{\mathcal{C}_{\mathrm{attn}}^{\mathrm{fine}}} = \frac{S_{\mathrm{amr}}^2}{S_f^2} = \frac{(1 + 3k)^2}{16}.$$

**Example ($k = 0.25$)** When $k = 0.25$, we have $1 + 3k = 1.75$, and thus

$$\frac{\mathcal{C}_{\mathrm{attn}}^{\mathrm{amr}}}{\mathcal{C}_{\mathrm{attn}}^{\mathrm{fine}}} = \frac{1.75^2}{16} = \frac{3.0625}{16} \approx 0.19.$$

That is, refining only $25\%$ of coarse patches reduces the self-attention computation to approximately $19\%$ of that required by full fine tokenization, corresponding to more than a $5\times$ reduction in the dominant attention cost.

This result highlights a key advantage of AMR-based tokenization: while full fine tokenization incurs a quadratic blow-up in sequence length, AMR controls the effective sequence length and yields a quadratic reduction in the dominant self-attention cost as a function of the refinement ratio $k$.

In addition to the above theoretical analysis, the following experiment can also demonstrate that our model can achieve a good balance between accuracy and efficiency.

We evaluate the accuracy–efficiency trade-off of spatial refinement across model scales. We consider three inference modes in a fixed order throughout this section: **No refinement** (uniform coarse tokens), **Ours** (AMR with 25% partial refinement), and **Full refinement** (uniform refinement everywhere). Accuracy is measured by the averaged relative $\ell_2$ error (Table 3), while efficiency is measured by the average forward runtime after warmup (Table 6).

*Table 6.* Average forward runtime (ms) for three model scales and refinement strategies.

| Model scale | No refinement | Ours | Full refinement |
|---|---|---|---|
| Small | 7.17 | 14.72 | 31.93 |
| Big | 18.04 | 39.74 | 106.93 |
| Large | 33.01 | 64.79 | 170.54 |

In addition to comparing refinement modes within MeshTok, we also report the mean inference time of MeshTok and representative baseline models under the same GPU environment, AMP setting, and inference protocol. Each model is run 50 times, and we report the average runtime in milliseconds.

*Table 7.* Inference-time comparison against baseline models. All models are evaluated under the same GPU environment, AMP setting, and inference protocol. Each model is run 50 times, and the mean runtime is reported in milliseconds. Lower runtime is better.

| Model | DeepONet | FNO | ViT | MPP | DPOT | MoE-POT | BCAT | Ours |
|---|---|---|---|---|---|---|---|---|
| Runtime (ms) $\downarrow$ | 44.78 | 30.63 | 7.40 | 90.41 | 17.11 | 50.03 | 17.79 | 39.74 |

Table 7 shows that MeshTok is not the fastest model in raw inference time, but it remains competitive with several Transformer-based PDE baselines while providing the accuracy gains reported in the main experiments. Thus, our runtime results should be interpreted as supporting an accuracy–efficiency trade-off rather than a claim of universally lowest latency.

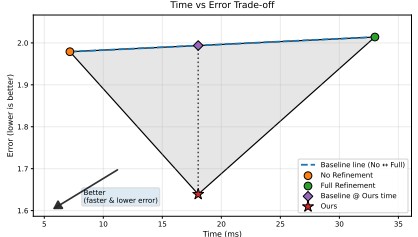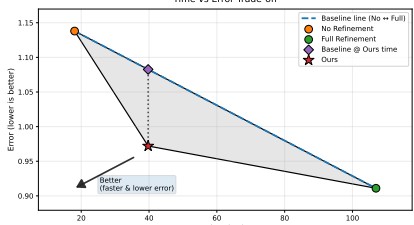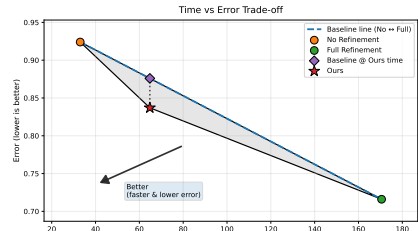

*Figure 6.* **Time–error trade-off across model scales.** Each subplot reports one-step relative $\ell_2$ error (y-axis, lower is better) versus average forward runtime after warmup (x-axis, in milliseconds). Subplots correspond to SMALL (left), BIG (middle), and LARGE (right). Within each subplot, the three operating points are ordered as *No refinement → Ours* (AMR 25%) *→ Full refinement*, where moving right increases compute and moving down improves accuracy. The figure illustrates how partial refinement shifts the operating point toward lower error with a moderate runtime increase, while avoiding the substantially larger overhead of uniform *Full refinement*.

Table 6 shows that refinement introduces a clear runtime hierarchy across all scales: *No refinement* is the fastest, *Full refinement* is the slowest, and *Ours (AMR 25%)* lies in between. For example, under BIG, AMR increases runtime from 18.04 ms to 39.74 ms, whereas *Full refinement* costs 106.93 ms per forward pass. A similar pattern holds for SMALL (7.17→14.72→31.93 ms) and LARGE (33.01→64.79→170.54 ms), indicating that partial refinement provides a moderate cost increase relative to the coarse baseline while remaining substantially cheaper than uniform *Full refinement*.

To directly connect runtime with accuracy, Figure 6 visualizes the operating points of the three modes for each model scale, where the $x$-axis is the average forward runtime and the $y$-axis is the one-step relative $\ell_2$ error (lower is better). Each subplot corresponds to one model scale (SMALL, BIG, LARGE), and each subplot contains three points ordered as *No refinement* → *Ours* → *Full refinement*. Thus, moving rightward indicates increased compute, while moving downward indicates improved accuracy.

Across all three scales, *Ours* consistently shifts the operating point downward compared to *No refinement*, demonstrating that partial refinement improves single-step accuracy with a moderate runtime increase. At the same time, *Ours* remains much closer to *No refinement* than to *Full refinement* in runtime, reflecting a more cost-efficient alternative to uniformly refining all patches.

The three subplots also reveal scale-dependent behavior. For SMALL, *Ours* improves accuracy over *No refinement* while avoiding the large overhead of *Full refinement*, and *Full refinement* is not always the best operating point in terms of one-step error under limited capacity (Table 3). For BIG, *Full refinement* achieves the lowest one-step error, but at a substantially higher runtime, whereas *Ours* provides a balanced point with reduced error relative to the coarse baseline and significantly lower cost than *Full refinement*. For LARGE, the gap between *Ours* and *Full refinement* in error becomes smaller, while the runtime gap remains large, suggesting that partial refinement remains attractive even when the backbone capacity is strong.

## B. Datasets

### B.1. Mathematical Formulations of the Datasets

We use four PDE benchmarks covering fluid dynamics, geophysical flows, and reaction–diffusion systems. For each dataset, we specify the quantity of interest (QoI), spatiotemporal domain, and the governing equations.

- **PDEBench: CNS$(\eta, \zeta)$ (Takamoto et al., 2022).** The QoI consists of the 2D compressible flow fields: velocity $\mathbf{u}(x,t) \in \mathbb{R}^2$, density $\rho(x,t)$, and pressure $p(x,t)$ over $(x,t) \in [0,1]^2 \times [0,T]$. Let $\eta$ denote the dynamic shear viscosity and $\zeta$ the bulk viscosity. The governing compressible Navier–Stokes equations are

$$\partial_t \rho + \nabla \cdot (\rho \mathbf{u}) = 0, \tag{6}$$

$$\rho\big(\partial_t \mathbf{u} + \mathbf{u} \cdot \nabla \mathbf{u}\big) = -\nabla p + \eta \Delta \mathbf{u} + \Big(\zeta + \tfrac{\eta}{3}\Big)\nabla(\nabla \cdot \mathbf{u}), \tag{7}$$

$$\partial_t E + \nabla \cdot \big((E+p)\mathbf{u}\big) = \nabla \cdot (\boldsymbol{\tau}\mathbf{u}), \tag{8}$$

where $E = \rho e + \frac{1}{2}\rho\|\mathbf{u}\|_2^2$ is the total energy density, $e$ is the internal energy per unit mass, and $\boldsymbol{\tau} = \eta\big(\nabla\mathbf{u} + (\nabla\mathbf{u})^\top - \frac{2}{3}(\nabla \cdot \mathbf{u})\mathbf{I}\big) + \zeta(\nabla \cdot \mathbf{u})\mathbf{I}$ is the viscous stress tensor. In our experiments, we follow the parameter settings and train/test splits from the original benchmark.

- **PDEBench: Shallow Water (Takamoto et al., 2022).** The QoI is the 2D shallow-water state $(h, \mathbf{v})$, where $h(x,t)$ is the fluid height and $\mathbf{v}(x,t) \in \mathbb{R}^2$ is the depth-averaged horizontal velocity on $(x,t) \in [0,1]^2 \times [0,T]$. The equations take the standard conservative form

$$\partial_t h + \nabla \cdot (h\mathbf{v}) = 0, \tag{9}$$

$$\partial_t(h\mathbf{v}) + \nabla \cdot \Big(h\mathbf{v} \otimes \mathbf{v} + \tfrac{1}{2}gh^2\mathbf{I}\Big) = \mathbf{S}(x,t), \tag{10}$$

where $g$ is the gravitational constant and $\mathbf{S}$ denotes external forcing and/or dissipation terms as specified by the dataset configuration.

- **PDEBench: Reaction–Diffusion (Takamoto et al., 2022).** The QoI consists of two coupled scalar fields $u(x,y,t)$ and $v(x,y,t)$ defined over a two-dimensional spatial domain $(x,y) \in [0,1]^2$ and time $t \in [0,T]$. The system is governed by a nonlinear reaction–diffusion process of the form

$$\partial_t u = D_u \Delta u + \mathcal{R}_u(u,v), \tag{11}$$

$$\partial_t v = D_v \Delta v + \mathcal{R}_v(u,v), \tag{12}$$

where $D_u$ and $D_v$ are diffusion coefficients, and $\mathcal{R}_u(\cdot)$ and $\mathcal{R}_v(\cdot)$ denote nonlinear reaction terms governing local interactions between the two species. This dataset exhibits rich spatiotemporal pattern formation driven by the interplay

between diffusion and nonlinear reactions, including the emergence of localized structures and sharp interfaces. The coexistence of smooth regions and fine-scale features makes this benchmark particularly challenging for learning models that must capture multi-scale spatial dynamics.

- **The Well: Gray–Scott (Ohana et al., 2024).** The QoI is a two-species concentration field $(u(x,t), v(x,t))$ over $(x,t) \in [0,1]^2 \times [0,T]$. The Gray–Scott reaction–diffusion system is

$$\partial_t u = D_u \Delta u - uv^2 + F(1-u), \tag{13}$$
$$\partial_t v = D_v \Delta v + uv^2 - (F+k)v, \tag{14}$$

where $D_u, D_v$ are diffusion coefficients, $F$ is the feed rate, and $k$ is the kill rate. This dataset is characterized by emergent fine-scale patterns and long-horizon dynamics.

- **The Well: shear flow (Ohana et al., 2024)** The QoI includes a 2D periodic incompressible velocity field $\mathbf{u}(t,x,z) = (u_x, u_z)$, pressure $p(t,x,z)$, and a passive tracer $s(t,x,z)$. The dynamics are governed by the incompressible Navier–Stokes equations

$$\partial_t \mathbf{u} - \nu \Delta \mathbf{u} + \nabla p = -\mathbf{u} \cdot \nabla \mathbf{u}, \tag{15}$$

together with an advection–diffusion equation for the tracer

$$\partial_t s - D \Delta s = -\mathbf{u} \cdot \nabla s, \tag{16}$$

where $\nu$ is the kinematic viscosity and $D$ is the tracer diffusivity. The system is parameterized by Reynolds and Schmidt numbers through $\nu = \mathrm{Re}^{-1}$ and $D = (\mathrm{Re} \cdot \mathrm{Sc})^{-1}$, and often exhibits instabilities (e.g., Kelvin–Helmholtz), vortex formation, and potential turbulent transition, requiring high resolution to capture fine-scale structures.

- **PDENNEval: Allen–Cahn (Wei et al., 2024).** The QoI is a scalar order parameter $u(x,t)$ on $(x,t) \in \Omega \times [0,T]$ (typically $\Omega \subset \mathbb{R}^2$ in the benchmark). The Allen–Cahn equation is

$$\partial_t u = \epsilon^2 \Delta u - (u^3 - u), \tag{17}$$

where $\epsilon > 0$ controls the interface width. The solutions exhibit moving interfaces and sharp transition layers, providing a stringent test for adaptive-resolution modeling.

- **PDENNEval: 2D Burgers (Wei et al., 2024).** The QoI is a two-component velocity field $\mathbf{u}(x,y,t) = (u(x,y,t), v(x,y,t))$ over $(x,y,t) \in [0,1]^2 \times [0,T]$. The 2D viscous Burgers system is

$$\partial_t u + u \, \partial_x u + v \, \partial_y u = \nu \left( \partial_{xx} u + \partial_{yy} u \right), \tag{18}$$
$$\partial_t v + u \, \partial_x v + v \, \partial_y v = \nu \left( \partial_{xx} v + \partial_{yy} v \right), \tag{19}$$

where $\nu$ is the viscosity (diffusion) coefficient. In PDENNEval, the Bu2 dataset is generated using a first-order upwind scheme for spatial first-order derivatives and a central difference scheme for second-order derivatives. This dataset is characterized by strong nonlinearity and shock-like steep gradients for small $\nu$, making it a challenging benchmark for learning long-horizon dynamics with sharp transients.

- **PDENNEval: Black–Scholes–Barenblatt Equation (Wei et al., 2024).** The Black–Scholes–Barenblatt equation is a nonlinear parabolic PDE arising in uncertain volatility modeling, defined as

$$\partial_t u + \frac{1}{2} \sigma^2(x) \, \partial_{xx} u + rx \, \partial_x u - ru = 0, \tag{20}$$

where $u(x,t)$ denotes the option price, $r$ is the risk-free interest rate, and $\sigma(x)$ is a state-dependent volatility function. This dataset features nonlinear diffusion effects and solution regimes with varying smoothness, serving as a challenging benchmark for general-purpose PDE learning models.

## B.2. Data Processing

All PDE datasets used in pretraining are transformed into a unified spatial resolution of $128 \times 128$, which is consistent with the majority of benchmarks considered in this work. For datasets with lower native resolution, we apply interpolation to upsample the solutions to $128 \times 128$, while higher-resolution data are downsampled via uniform subsampling. This normalization enables joint training across heterogeneous datasets drawn from PDEBench (Takamoto et al., 2022), PDENNEval (Wei et al., 2024), and The Well (Ohana et al., 2024).

To accommodate varying numbers of physical variables across different equations, we pad the channel dimension of all states to four channels, noting that nearly all PDE systems in the considered benchmarks have at most four state variables. Channel-wise masking is applied to ensure that padded dimensions do not contribute to the loss.

To improve generalization and mitigate the accumulation of errors during autoregressive rollout, we inject additive noise into the training data. The noise magnitude is scaled proportionally to the data amplitude, with a standard deviation set to $1\%$ of the corresponding signal magnitude, following common practice in data-driven PDE modeling.

Finally, to ensure balanced learning across different equation types, all PDE systems are sampled with equal probability during training, preventing domination by any single dataset or equation family.

## C. The Details of the Model

**Ours**  We consider three model scales—*Small*, *Medium*, and *Large*—to study the impact of model capacity and spatial resolution. All models share the same overall architecture and differ only in depth, embedding dimension, feed-forward width, and patch resolution.

- **Small model.** The SMALL configuration uses a lightweight Transformer with 4 layers, attention dimension $d = 256$, and MLP dimension $d_{\text{ff}} = 512$. We use 8 attention heads and a fixed spatial patch size of $16 \times 16$, corresponding to an $8 \times 8$ coarse token grid for $128 \times 128$ inputs. This model serves as the low-capacity setting in our scaling study.

- **Big model.** The BIG configuration increases the backbone capacity to 8 layers with attention dimension $d = 512$ and MLP dimension $d_{\text{ff}} = 1280$, while keeping the number of heads fixed at 8. The spatial tokenization remains unchanged with the same $16 \times 16$ patch size (i.e., an $8 \times 8$ token grid), so performance differences primarily reflect the effect of scaling the Transformer backbone.

- **Large model.** The LARGE configuration further scales the Transformer to 12 layers with attention dimension $d = 768$ and MLP dimension $d_{\text{ff}} = 1600$, again using 8 attention heads. We keep the same $16 \times 16$ patch size for spatial tokenization, ensuring that comparisons across scales isolate capacity changes in the Transformer. This is the highest-capacity configuration evaluated in our experiments.

Across all scales, we use 8 attention heads, RMS normalization, SwiGLU activation, convolutional patch embedding, and learnable time embeddings, ensuring that performance differences primarily arise from model scale rather than architectural changes. In total, the specific configuration is shown in the table 8.

*Table 8.* Model configurations at different scales.

| Size | Attention dim | MLP dim | Layers | Heads | Patch size |
|------|--------------|---------|--------|-------|------------|
| Small | 256 | 512 | 4 | 8 | $16 \times 16$ |
| Big | 512 | 1280 | 8 | 8 | $16 \times 16$ |
| Large | 768 | 1600 | 12 | 8 | $16 \times 16$ |

**Baseline Configurations**  All baseline models are configured to have model capacity comparable to our *Big* configuration, ensuring a fair comparison in terms of parameter count and representational power. Unless otherwise specified, architectural choices and hyperparameters follow standard implementations for each baseline, with key dimensions adjusted to closely match our Big model.

- **DeepONet** The DeepONet baseline employs a dual-branch operator learning architecture composed of a branch network and a trunk network. The branch network uses a basis dimension of 256 to encode input function samples, while the trunk network uses an embedding dimension of 512 to represent spatial coordinates. The model adopts a two-branch design (i.e., `singlebranch = false`) and maps inputs defined on an $16 \times 16$ patch grid to a single output patch.

- **FNO** The Fourier Neural Operator baseline adopts a spectral convolution-based operator learning architecture. It retains 8 Fourier modes along each spatial dimension (height, width, and depth) and uses a hidden channel width of 64. The model consists of 8 stacked spectral convolution layers and operates directly on the full spatial grid, without relying on patch-based tokenization.

- **ViT** The Vision Transformer baseline follows a standard encoder-only Transformer architecture. It uses an embedding dimension of 512 and a feed-forward network dimension of 2048. The input field is tokenized into $16 \times 16$ non-overlapping patches, resulting in a fixed-length token sequence for Transformer processing. The same patch resolution is used for both input and output representations.

- **DPOT** DPOT adopts a patch-based operator Transformer with Fourier-domain mixing. We use a patch size of $16 \times 16$ on $128 \times 128$ inputs, resulting in an $8 \times 8$ latent grid. The model employs an embedding dimension of 512 and a depth of 8 Transformer-style blocks. Each block consists of AFNO-based spectral mixing followed by a feed-forward module. Temporal dependencies are handled via a dedicated temporal aggregation layer, and the number of retained Fourier modes is set to 32.

- **MPP** MPP is instantiated using an axial space–time Transformer architecture. For a $128 \times 128$ input field, we tokenize the spatial domain into non-overlapping patches of size $16 \times 16$, yielding an $8 \times 8$ patch (token) grid per frame. Each patch is mapped to an embedding of dimension 512. The model stacks 8 space–time processing blocks, where each block alternates between temporal self-attention and axial spatial attention along horizontal and vertical directions. A lightweight MLP is used for channel-wise mixing, and relative or continuous positional biases are applied within attention modules.

- **MoE-POT** MoE-POT extends operator Transformers with mixture-of-experts feed-forward layers. We configure the model with an embedding dimension of 512, a depth of 8 blocks, and 16 experts per MoE layer, with top-$k$ routing enabled. The patch resolution is $16 \times 16$, consistent with other baselines. Spectral mixing is implemented via AFNO modules, while expert routing dynamically allocates computation based on global feature statistics.

- **BCAT** BCAT is a Transformer-based operator learning framework that introduces several architectural innovations tailored for spatiotemporal PDE modeling. In our experiments, BCAT is configured with an $16 \times 16$ patch resolution, an embedding dimension of 512, and a depth of 8 Transformer blocks, yielding a model capacity comparable to our *Big* configuration.

## D. Training and Evaluation Settings

**Training Settings**  Autoregressive training has been widely used in language and image generation, where each prediction is conditioned on previously generated or observed context (Bengio et al., 2000; Brown et al., 2020; van den Oord et al., 2016). We transferred it to the PDE field. We adopt a unified training protocol for all experiments to ensure stable autoregressive learning and fair comparison across PDE benchmarks. Each training example is a spatiotemporal trajectory

$$x \in \mathbb{R}^{T \times H \times W \times C},$$

where $T$ is the number of time steps, and $H \times W$ denotes the spatial resolution with $C$ physical channels. We use an input history length of $T_{\mathrm{in}} = 10$ (`input_len=10`) and train the model in an autoregressive next-step prediction manner.

**Autoregressive Training Objective**  Let $f_\theta$ denote the model with parameters $\theta$. Given an input history window $\mathbf{x}_{t-T_{\mathrm{in}}+1:t} = \{x^{t-T_{\mathrm{in}}+1}, \ldots, x^t\}$, the one-step prediction is defined as

$$\hat{x}^{t+1} = f_\theta\big(x^{t-T_{\mathrm{in}}+1}, \, x^{t-T_{\mathrm{in}}+2}, \, \ldots, \, x^t\big).$$

During training, the loss is computed only on the predicted portion of the sequence (`loss_start_idx=`$T_{\text{in}}$), i.e., we do not penalize the observed history. Using mean-squared error in the normalized space, the per-step training loss is

$$\mathcal{L}_{t+1} = \left\| \hat{x}^{t+1} - x^{t+1} \right\|_2^2,$$

where $\| \cdot \|_2$ denotes the Euclidean norm over all spatial locations and channels.

**Normalization**   To reduce scale variation across PDE families and stabilize optimization, we apply mean–variance normalization (`normalize=meanvar`) (Ba et al., 2016). Let $\mu$ and $\sigma$ denote the dataset (or batch) mean and standard deviation. We normalize inputs as

$$\tilde{x} = \frac{x - \mu}{\sigma},$$

and compute the training objective in the normalized space (`denormalize_for_loss=0`). This improves numerical conditioning and prevents a small subset of variables from dominating the loss.

**Noise Augmentation**   We augment training trajectories with *relative* additive noise of strength $\delta = 0.01$ (`noise=0.01`, `noise_type=additive`). Unlike using a fixed absolute noise scale, our perturbation magnitude is proportional to the norm of the underlying field, making the augmentation scale-invariant across PDEs with different physical units and amplitudes. Concretely, for each state $x^t \in \mathbb{R}^{H \times W \times C}$, we construct a noisy input

$$\tilde{x}^t = x^t + \delta \, \|x^t\|_2 \cdot \xi^t, \qquad \xi^t \sim \mathcal{N}(0, \mathbf{I}),$$

where $\| \cdot \|_2$ denotes the Euclidean norm over all spatial locations and channels, and $\xi^t$ is i.i.d. Gaussian noise with unit variance. This relative perturbation improves robustness to distribution shifts across equations and regimes, and empirically enhances rollout stability by exposing the model to slightly off-manifold states during training, thereby partially mitigating compounding errors in long-horizon autoregressive inference (Bishop, 1995; Srivastava et al., 2014; Vincent et al., 2008; Neelakantan et al., 2015).

**Optimizer**   We train with AdamW (decoupled weight decay) (Loshchilov & Hutter, 2017) using learning rate $\eta_0 = 10^{-4}$, weight decay $10^{-4}$, and numerical stabilization $\epsilon = 10^{-6}$. We set $\beta_1 = 0.9$ and $\beta_2 = 0.95$ (`beta2=0.95`) and disable AMSGrad (`amsgrad=false`). The choice $\beta_1 = 0.9$ provides stable first-moment smoothing under stochastic gradients, while a moderately smaller $\beta_2$ (relative to the common 0.999) makes the second-moment estimate more responsive to non-stationary gradient statistics frequently observed in autoregressive sequence learning and multi-dataset PDE training, thereby improving adaptation without sacrificing stability. We additionally apply gradient clipping with a global norm threshold of 1.0 (`clip_grad_norm=1.0`) to prevent occasional gradient spikes.

**Cosine Schedule with Warmup**   We train for $E = 20$ epochs with $S = 4000$ optimization steps per epoch, yielding the total number of optimizer updates

$$I_{\max} = E \cdot S = 20 \times 4000 = 80000,$$

which corresponds to `max_iters`. We use a cosine learning-rate schedule (`scheduler_type=cosine`) with linear warmup occupying a fraction $w = 0.1$ of all iterations (`warmup=0.1`) (Loshchilov & Hutter, 2016; Goyal et al., 2017). Let

$$I_{\text{w}} = \lfloor w I_{\max} \rfloor$$

be the warmup length, and let $i \in \{0, 1, \dots, I_{\max}\}$ denote the global optimization step. The learning rate $\eta(i)$ is defined by the following piecewise function:

$$\eta(i) = \begin{cases} \eta_0 \cdot \dfrac{i}{I_{\text{w}}}, & 0 \le i \le I_{\text{w}}, \\ \eta_0 \cdot \dfrac{1}{2} \left( 1 + \cos\left( \pi \dfrac{i - I_{\text{w}}}{I_{\max} - I_{\text{w}}} \right) \right), & I_{\text{w}} < i \le I_{\max}. \end{cases}$$

Warmup avoids unstable updates before the moment estimates become reliable, while cosine decay smoothly anneals the learning rate for convergence.

**Mixed Precision and Batch Size**   We use a batch size of $8$ (batch_size=8) and enable automatic mixed precision (amp=1) to accelerate training and reduce memory footprint, while preserving stable optimization dynamics.

**Autoregressive Rollout Evaluation**   Beyond one-step prediction, we evaluate models under closed-loop autoregressive rollout to assess long-horizon stability. Let the normalized $\ell_2$ error at horizon $h$ be

$$\text{Err}_h = \frac{\left\|\hat{x}^{t+h} - x^{t+h}\right\|_2}{\left\|x^{t+h}\right\|_2}.$$

We focus on two representative horizons, $h = 1$ and $h = 10$, which respectively reflect (i) short-horizon fitting ability and (ii) robustness to compounding errors under rollout. For $h = 1$, $\hat{x}^{t+1}$ is computed from the ground-truth history as defined above. For $h > 1$, predictions are recursively fed back as inputs. Specifically, for $h \geq 2$ we define

$$\hat{x}^{t+h} = f_\theta\big(\hat{x}^{t+h-T_{\text{in}}}, \hat{x}^{t+h-T_{\text{in}}+1}, \ldots, \hat{x}^{t+h-1}\big),$$

where the first $T_{\text{in}}$ states are initialized using ground-truth observations, and subsequent states are model predictions. This recursion makes explicit the distribution shift from teacher-forced training to free-running inference, and directly measures how prediction errors accumulate over time.

For completeness, we also report an integrated rollout error over a horizon window of length $H$ (corresponding to _l2_error_int),

$$\text{Err}_{\text{int}} = \frac{1}{H} \sum_{h=1}^{H} \frac{\left\|\hat{x}^{t+h} - x^{t+h}\right\|_2}{\left\|x^{t+h}\right\|_2}.$$

Model selection uses validation $\ell_2$ error (validation_metrics=_l2_error), and we report both $\text{Err}_1$ and $\text{Err}_{10}$ to characterize predictive accuracy and rollout stability.

**Equation-Specific Fine-Tuning**   For downstream equation-specific adaptation, we fine-tune the pretrained model for $E = 5$ epochs with $S = 4000$ optimization steps per epoch, resulting in a total of

$$I_{\max} = E \cdot S = 5 \times 4000 = 20000$$

parameter updates. All other training settings are kept identical to pretraining (normalization, relative noise augmentation, batch size, mixed precision, and gradient clipping) to isolate the effect of initialization and fine-tuning itself. We optimize with AdamW using learning rate $\eta_0 = 10^{-4}$, weight decay $10^{-4}$, $\epsilon = 10^{-6}$, and momentum parameters $(\beta_1, \beta_2) = (0.9, 0.95)$. The learning rate follows a cosine schedule with linear warmup ratio $w = 0.1$, where the warmup length is

$$I_{\text{w}} = \lfloor wI_{\max} \rfloor = \lfloor 0.1 \times 20000 \rfloor = 2000,$$

and for global step $i \in \{0, \ldots, I_{\max}\}$ the learning rate is

$$\eta(i) = \begin{cases} \eta_0 \cdot \dfrac{i}{I_{\text{w}}}, & 0 \leq i \leq I_{\text{w}}, \\ \eta_0 \cdot \dfrac{1}{2}\left(1 + \cos\left(\pi \dfrac{i - I_{\text{w}}}{I_{\max} - I_{\text{w}}}\right)\right), & I_{\text{w}} < i \leq I_{\max}. \end{cases}$$

We evaluate both short-horizon fitting ability and long-horizon stability using rollout errors at horizons $h \in \{1, 10\}$,

$$\text{Err}_h = \frac{\left\|\hat{x}^{t+h} - x^{t+h}\right\|_2}{\left\|x^{t+h}\right\|_2},$$

where $\hat{x}^{t+1} = f_\theta(x^{t-9}, \ldots, x^t)$ and multi-step predictions are obtained recursively by feeding back model outputs. During fine-tuning, we compare two initialization strategies: training *from scratch* with random initialization versus *warm-start fine-tuning* initialized from the final pretrained checkpoint, quantifying the benefit of foundation-model pretraining for downstream PDE adaptation.

*Table 9.* Averaged relative $\ell_2$ rollout error for the TNT-style refinement variant (lower is better).

| Model | 1-step | 5-step | 10-step |
|---|---|---|---|
| TNT-style | 1.050 | 1.708 | 2.461 |
| Ours | 0.972 | 1.663 | 2.261 |

# E. Architecture Exploration: Alternative Token Refinement Mechanisms

Our main architecture performs adaptive multi-scale modeling by directly mixing coarse and refined tokens in a *single* block-causal Transformer, enabling cross-resolution interactions throughout all layers. There are also other ways to handle multi-scale tokens (Chen et al., 2021; 2023; Wang et al., 2021; Li et al., 2022b; Guibas et al., 2021). To better understand the design space of refinement mechanisms, we additionally investigate two alternative schemes: (i) a Transformer-in-Transformer refinement inspired by TNT, and (ii) a physics-aware token formation strategy inspired by Transolver. In both explorations, we keep the same refinement indicator and enforce block-causal self-attention along the temporal dimension.

## E.1. TNT-Style Local Refinement with Global Feature Injection

**Motivation and Formulation** Transformer-in-Transformer (TNT) (Han et al., 2021) is originally introduced to address a key limitation of patch-based Transformers: when the backbone operates on relatively coarse patches, fine-scale patterns within a patch may be under-represented, since the global Transformer primarily models interactions *between* patch tokens rather than *within* a patch. TNT resolves this by allocating dedicated modeling capacity to local structures through a secondary Transformer operating on sub-patch tokens, and then injecting the resulting local representation into the corresponding coarse token before global processing. This design is conceptually aligned with AMR: refined regions naturally deserve additional computation to capture sharp gradients, vortical patterns, or stiff reaction fronts.

**Model Design** We adapt TNT to our PDE forecasting setup with the same activity-based refinement policy used in MeshTok. Starting from a coarse partition, the indicator selects a subset of coarse patches that require refinement. For each selected coarse patch, we further subdivide it into $m$ sub-patches and embed them as local tokens $\{\mathbf{u}_{i,j}\}_{j=1}^m$. A lightweight *local Transformer* $\text{LocalTrans}(\cdot)$ then computes a refined summary feature $\mathbf{r}_i$ that captures high-frequency behaviors and intra-patch dependencies:

$$\mathbf{r}_i = \text{LocalTrans}\big(\{\mathbf{u}_{i,j}\}_{j=1}^m\big).$$

The refined feature is fused back into the coarse token $\mathbf{z}_i$ via a feature-injection operator, e.g., additive residual or concatenation followed by projection:

$$\tilde{\mathbf{z}}_i = \text{Fuse}(\mathbf{z}_i, \mathbf{r}_i).$$

All coarse tokens (refined or not) form a global token sequence that is processed by a *global* Transformer backbone with block-causal temporal attention:

$$\widetilde{\mathbf{Z}} = \text{GlobalTrans}_{\text{BC}}(\{\tilde{\mathbf{z}}_i\}_i).$$

Here, block-causality enforces that tokens at future time steps attend only to tokens from the past, while spatial attention remains fully dense within the same time step. Notably, this TNT-style hierarchy introduces a two-stage information flow (local encoding $\rightarrow$ injection $\rightarrow$ global modeling), which provides a strong baseline since refined patches are guaranteed to receive additional local modeling capacity before global integration.

**Comparison Protocol** We compare TNT-style refinement against our unified AMR token mixing under identical data splits, training schedules, and evaluation metrics. All models use the same indicator selection and the same block-causal temporal attention constraint. Performance is measured via normalized rollout $\ell_2$ errors at horizons $h \in \{1, 5, 10\}$ (lower is better), which is shown in Table 9.

**Analysis** The TNT-style baseline yields reasonably competitive results, which is expected given its explicit allocation of local refinement capacity for selected patches. However, our approach remains consistently better across short and long horizons. We attribute this to the fact that TNT enforces a hierarchical separation between local and global processing: refined features are first compressed by a local Transformer and then injected into coarse tokens, after which the global backbone models only the coarse-level token sequence. This compression-and-injection step can bottleneck fine-scale

*Table 10.* Averaged relative $\ell_2$ rollout error for the Transolver-style token formation variant (lower is better).

| Model | 1-step | 5-step | 10-step |
|---|---|---|---|
| Transolver-style | 5.273 | 7.353 | 8.099 |
| Ours | 0.972 | 1.663 | 2.261 |

information and introduce representational mismatch between local summaries and global token semantics. In contrast, our model directly places coarse and refined tokens into a single unified sequence and allows cross-resolution attention at *every* layer, enabling fine-scale details to interact with global context without an explicit bottleneck. Importantly, our unified design achieves higher accuracy with less architectural overhead, since it avoids maintaining a dedicated local Transformer and instead leverages shared capacity for both local and global reasoning.

### E.2. Transolver-Style Physics-Aware Token Formation via Soft Clustering

**Motivation and Formulation**   Transolver (Wu et al., 2024) advocates a physics-inspired attention mechanism that does not rely on rigid, axis-aligned patch partitioning. The key observation is that physically meaningful structures in PDE solutions (e.g., shocks, shear layers, mixing interfaces) may not align with a uniform patch grid. Instead of treating each spatial patch as a token, Transolver forms tokens through a *soft clustering* process over feature space, where points (or small cells) with similar physical states are aggregated into latent tokens. This tokenization is potentially appealing for AMR-style representations, as AMR also produces non-uniform discretizations with heterogeneous cell sizes and refinement depths. We therefore hypothesize that physics-aware grouping might complement non-uniform token layouts beyond purely geometric patching.

**Model Design**   We instantiate a Transolver-style token formation module under our forecasting setting. Given pointwise features $\{\mathbf{h}_n\}_{n=1}^{N}$ extracted from the input field, we compute soft assignment weights to $K$ latent prototypes $\{\mathbf{c}_k\}_{k=1}^{K}$:

$$\alpha_{n,k} = \mathrm{SoftAssign}(\mathbf{h}_n, \mathbf{c}_k), \qquad \sum_{k=1}^{K} \alpha_{n,k} = 1, \; \alpha_{n,k} \geq 0.$$

Latent tokens are then aggregated by weighted pooling:

$$\mathbf{v}_k = \sum_{n=1}^{N} \alpha_{n,k} \, \mathbf{h}_n.$$

The resulting token set $\{\mathbf{v}_k\}_{k=1}^{K}$ is fed into a Transformer backbone with block-causal temporal attention for autoregressive prediction. Crucially, to ensure fairness, we do *not* allow Transolver to gain an advantage simply by changing the refinement source: all tokens originate from the same physics-aware refinement setting used by our method (rather than uniform spatial patches), and the difference lies only in whether the tokens are kept in a structured multi-scale layout (*Ours*) or re-formed by soft clustering (Transolver-style).

**Comparison Protocol**   We compare this Transolver-style token formation against our unified MeshTok under identical optimization settings and rollout metrics. Evaluation uses the same normalized rollout $\ell_2$ errors at horizons $h \in \{1, 5, 10\}$, which is shown in Table 10

**Analysis**   The Transolver-style performs substantially worse in our setting. A plausible explanation is a mismatch between soft clustering and the refinement-aware multi-scale tokenization prior. Our method constructs tokens with explicit geometric and hierarchical semantics: each token corresponds to a fixed spatial region with a well-defined refinement depth, and the sequence ordering preserves locality and scale in a structured manner. Such a prior is particularly useful for PDE forecasting, where long-horizon rollout benefits from consistent spatiotemporal coupling and predictable information propagation across scales. In contrast, soft clustering introduces an adaptive *soft assignment* that aggregates features across spatial locations based on instantaneous similarity. While flexible, this data-dependent mixing can reduce the locality and scale consistency implied by the AMR partition, and may blur sharp interfaces or localized structures. As a result, the learned aggregation may interfere with the intended refinement-aware positional and multi-scale representation, which can complicate optimization

and reduce long-horizon stability. Therefore, although physics-aware clustering is appealing in general, in our AMR-guided setting, it may not align with the structured discretization prior already encoded by the coarse–fine tokenization.

# F. More Ablation Studies

## F.1. Seed Robustness for Refinement Scaling

To assess whether the scaling comparison is sensitive to random initialization, we repeat the main refinement scaling study with three independent seeds while keeping the data splits, model configurations, and training protocol unchanged.

*Table 11.* Three-seed robustness study for the main refinement scaling comparison. We report mean $\pm$ standard deviation of relative $\ell_2$ error. Lower is better.

| Scale | No refinement | AMR | Full refinement |
|-------|---------------|-----|-----------------|
| Small | $2.013 \pm 0.031$ | $\mathbf{1.631 \pm 0.009}$ | $1.933 \pm 0.071$ |
| Big | $1.122 \pm 0.015$ | $0.977 \pm 0.007$ | $\mathbf{0.914 \pm 0.009}$ |
| Large | $0.917 \pm 0.007$ | $0.842 \pm 0.006$ | $\mathbf{0.727 \pm 0.016}$ |

Across all three model scales in this robustness study, AMR consistently improves over no refinement with small standard deviations across seeds. Full refinement achieves the lowest error for the BIG and LARGE configurations, but it uses a substantially larger token sequence and higher runtime, as discussed in the main scaling and efficiency results. These results support the conclusion that the observed AMR gains over coarse tokenization are not caused by a single favorable random seed.

## F.2. Indicator-Guided Refinement and Refinement Policies

MeshTok allocates extra spatial resolution via *indicator-guided patch refinement*: under a fixed refinement budget, only a subset of coarse patches is refined into finer sub-patches. In this ablation, we compare four refinement mechanisms that differ in how they select refined regions: **activity-based** refinement, **random** refinement, **a posteriori error-estimation** refinement, and **end-to-end** refinement. All mechanisms refine the same number of coarse patches and use the same downstream predictor; only the refinement policy is changed.

Let $x_t \in \mathbb{R}^{H \times W \times C}$ denote the PDE field at time $t$. We partition the domain into $N$ non-overlapping coarse patches $\{\mathcal{P}_i\}_{i=1}^N$ of size $p \times p$ (we use $p = 8$), and top-$k$ selection is performed independently per sample (and per time step), enforcing exactly $k$ refined patches for each frame. Each policy produces a refinement mask $m_t \in \{0,1\}^N$ satisfying $\sum_{i=1}^N m_t(i) = k$, where $m_t(i) = 1$ indicates that patch $i$ is refined.

**Activity-Based Refinement** Activity-based refinement assigns each patch a physics-motivated score derived directly from the current field. A typical instantiation aggregates local gradients and energy within each coarse patch:

$$s_i^{\text{act}}(x_t) = \frac{1}{|\mathcal{P}_i|} \sum_{u \in \mathcal{P}_i} \Big( \|\nabla x_t(u)\|_2 + \lambda \|\Delta x_t(u)\|_2^2 \Big),$$

where $\nabla$ denotes discrete spatial gradients and $\lambda$ balances the two terms. Refinement is triggered by selecting the top-$k$ patches with the largest $s_i^{\text{act}}$.

**Random refinement.** Random refinement selects $k$ patches uniformly at random and refines them, without using any solution-dependent information. While simple, it can be interpreted as a strong stochastic augmentation of the tokenization process, since the predictor is trained under diverse refinement patterns.

**A posteriori error-estimation refinement.** A posteriori refinement selects patches according to their estimated refinement benefit. This variant is used only as an ablation to study indicator quality: it does not use the true future solution error at inference time, since such information is unavailable during deployment and would constitute an oracle signal. To construct supervision for a lightweight indicator, we compare two predictors evaluated on the same input field $x_t$: a coarse-only predictor $f_{\text{coarse}}$ (*No refinement*) and a fully refined predictor $f_{\text{full}}$ (all patches refined). Let $\ell_{\text{coarse}}(u,c)$ and $\ell_{\text{fine}}(u,c)$ denote the per-location, per-channel loss contributions at spatial location $u$ and variable channel $c$ (we use relative $\ell_2$ loss in our

experiments). We define a non-negative improvement signal

$$\Delta\ell(u, c) = \max\big(\ell_{\text{coarse}}(u, c) - \ell_{\text{fine}}(u, c),\, 0\big),$$

and reduce it across channels to obtain a scalar improvement field,

$$\Delta\ell_s(u) = \text{Reduce}_c\big(\Delta\ell(u, c)\big),$$

where $\text{Reduce}_c$ is implemented as either mean or sum. For each coarse patch $\mathcal{P}_i$, we compute the patch-level improvement score by averaging within the patch,

$$s_i^{\text{post}}(x_t) = \frac{1}{|\mathcal{P}_i|} \sum_{u \in \mathcal{P}_i} \Delta\ell_s(u).$$

Given a refinement budget $k$, we convert these scores into binary refinement labels via top-$k$ selection:

$$y_i(x_t) = \mathbf{1}\Big(i \in \text{TopK}(\{s_j^{\text{post}}(x_t)\}_{j=1}^N,\, k)\Big) \in \{0, 1\}.$$

We then train a lightweight indicator network $g_\phi$ (a 3-layer U-Net with hidden width 32) to predict this binary refinement mask from $x_t$ using a classification objective. Specifically, we minimize the binary cross-entropy loss

$$\phi^\star = \arg\min_\phi\ \mathbb{E}\Big[\text{BCE}\big(\sigma(g_\phi(x_t)),\, y(x_t)\big)\Big],$$

where $\sigma(\cdot)$ denotes the sigmoid function. We train $g_\phi$ for 10,000 iterations with batch size 64, and then freeze $g_{\phi^\star}$ for refinement. During refinement, the indicator requires only a single forward pass to produce refinement scores, and the final refined set is obtained by selecting the top-$k$ predicted patches. Therefore, the two-predictor comparison is only used offline to generate training labels for the indicator and is not required once the indicator has been trained. We do not use the true solution error itself as an inference-time refinement signal, since it is unavailable during deployment and would constitute an oracle indicator.

**End-to-end refinement.** End-to-end refinement learns the indicator jointly with the downstream predictor via the final prediction loss, treating refinement as part of the overall network. In practice, we find this training procedure unstable: the learned indicator often collapses to nearly input-independent refinement maps and yields limited adaptivity. We therefore report this degenerated end-to-end variant as *Model (fixed)* in Table 12.

To ensure a fair comparison, we keep the predictor architecture, refinement budget $k$, optimizer, and training schedule fixed, and vary only the refinement policy. We report the averaged relative $\ell_2$ error for **1-step** prediction and **10-step** rollout, where the 10-step rollout is obtained by iteratively feeding the model's own predictions back as inputs. Errors are averaged over evaluation trajectories and time steps, producing the summary in Table 12.

*Table 12.* Averaged relative $\ell_2$ error (lower is better) under different refinement strategies.

| | | Refinement strategy | | |
|---|---|---|---|---|
| Horizon | Random | Model (fixed) | Activity-based | A posteriori-based |
| 1-step | 1.080 | 1.068 | 0.972 | **0.967** |
| 10-step | 2.311 | 2.521 | **2.261** | 2.272 |

Table 12 shows that both activity-based and a posteriori-based refinement improve short-horizon accuracy over random refinement (0.972/0.967 vs. 1.080 at 1-step). For the 10-step rollout, activity-based refinement attains the lowest error (2.261), while a posteriori-based refinement remains close (2.272), and random refinement is higher (2.311). Table 12 also reports *Model (fixed)* to quantify the degenerated behavior of the end-to-end learned indicator. Although this variant obtains a moderate 1-step error of 1.068, its 10-step rollout error rises to 2.521. This gap indicates that a nearly input-independent refinement pattern is brittle under autoregressive dynamics: as high-activity regions move over time, stale refinement locations can lead to accumulating errors.

Overall, activity-based and a posteriori-based refinement both provide strong accuracy under the same refinement budget, with a posteriori-based refinement being slightly better for 1-step prediction and activity-based refinement being slightly

better for 10-step rollout. The main practical difference is complexity and runtime: activity-based refinement is simple to compute from $x_t$, whereas a posteriori refinement requires training and deploying an additional indicator network, and its end-to-end runtime in our implementation reaches roughly 70% of the cost of *Full refinement*. Based on this trade-off between accuracy and efficiency, we adopt activity-based refinement as the default policy in the main experiments.

### F.3. Refinement Ratio Study

We further study the refinement ratio $k$ through the lens of classical adaptive mesh refinement (AMR). In numerical PDE solvers, refinement can be interpreted as allocating additional degrees of freedom to reduce *local truncation error*, thereby better resolving fine-scale structures such as sharp gradients, thin shear layers, or stiff reaction fronts. In our formulation, $k \in \{0.125, 0.25, 0.375\}$ specifies the fraction of coarse patches refined at each time step. Concretely, given per-patch scores produced by the indicator, we perform a top-$k$ selection independently for each sample (and each time step), enforcing that exactly $k$ of the coarse patches are refined in every frame.

Operationally, refining a patch replaces one coarse token with a small set of fine tokens, increasing the local spatial sampling density and expanding the sequence length. If the coarse token grid contains $N$ patches and each refined patch is split into $s$ fine tokens (e.g., $s = 4$ for a $2 \times 2$ split), then the sequence length grows from $N$ to

$$L(k) \approx N + kN(s - 1),$$

which scales approximately linearly with $k$. Therefore, $k$ provides a controlled knob that trades computation for accuracy by directing additional modeling capacity to the regions predicted to be most difficult.

From a numerical analysis viewpoint, coarse discretizations incur larger local truncation errors when the solution contains high-frequency content, e.g., shocks, sharp interfaces, or rapidly varying patterns. Under-resolution in such regions can manifest as smoothing, phase errors, or aliasing artifacts, and these local inaccuracies are often the dominant contributors to global prediction error. Analogously, our refinement increases the effective local resolution: fine tokens represent smaller spatial regions and thus can preserve sharper spatial variations with less information loss at tokenization time. This typically improves one-step forecasting by reducing local modeling bias in the most challenging regions.

More importantly, refinement can have a larger impact on long-horizon autoregressive rollout. Even if the one-step error is moderate, rollout repeatedly feeds predictions back as inputs, so local errors can propagate and accumulate through the underlying dynamics. This aligns with classical error-propagation intuition in time-marching schemes, where the horizon-$h$ error reflects both the instantaneous approximation error and its amplification by the evolution operator. As a result, refining the primary "error-producing" regions can yield a disproportionately large improvement in multi-step stability.

To quantify this effect, we train the same model under three refinement ratios $k \in \{0.125, 0.25, 0.375\}$ while keeping all other settings fixed (data splits, optimization hyperparameters, and training budget). For evaluation, we compute the relative $\ell_2$ error for (i) one-step prediction and (ii) 10-step autoregressive rollout, and then average the errors over all benchmark datasets. The resulting averaged errors are summarized in Table 13 (lower is better).

*Table 13.* Averaged relative $\ell_2$ rollout error under different refinement ratios $k$ (lower is better).

| Horizon | Refinement ratio $k$ | | |
|---|---|---|---|
| | 0.125 | 0.25 | 0.375 |
| **1-step** | 1.026 | 0.972 | 0.963 |
| **10-step** | 2.476 | 2.261 | 2.254 |

Table 13 shows that increasing $k$ consistently improves both 1-step and 10-step accuracy. For one-step prediction, the error decreases from 1.026 at $k = 0.125$ to 0.972 at $k = 0.25$ and further to 0.963 at $k = 0.375$. For the 10-step rollout, the reduction is larger in magnitude, dropping from 2.476 to 2.261 and then to 2.254 as $k$ increases. These results suggest that a higher refinement ratio helps suppress local errors in difficult regions and reduces their accumulation over long horizons.

Although $k = 0.375$ achieves the lowest errors, we adopt $k = 0.25$ as the default setting due to its accuracy–efficiency balance. A larger $k$ increases the token count and thus the attention cost, which grows super-linearly with sequence length. Empirically, $k = 0.25$ already captures most of the accuracy gains observed at $k = 0.375$, while avoiding the additional overhead of more aggressive refinement. Unless otherwise specified, we therefore use $k = 0.25$ in the main experiments.

### F.4. Resolution Generalization

Most experiments in this work are conducted at a unified spatial resolution of $128 \times 128$ to enable joint training and fair comparison across heterogeneous PDE benchmarks. To further examine whether our refinement-aware tokenization remains effective when the spatial resolution increases, we perform an additional ablation at $256 \times 256$.

All methods in this ablation use the same BIG configuration, and differ only in the refinement mode. We construct $256 \times 256$ inputs by applying bilinear interpolation to upsample the original $128 \times 128$ fields, and keep all other evaluation settings unchanged. We compare three refinement modes: *(i) No refinement*, which uses only coarse tokens with a uniform coarse partition; *(ii) Ours (AMR 25%)*, which refines a fixed fraction of patches selected by the indicator at each time step; and *(iii) Full refinement*, which uniformly applies the finest tokenization across the entire domain. Performance is measured by relative $\ell_2$ error (lower is better), and we additionally report an overall **Avg.** score obtained by averaging the errors across the five PDEs.

*Table 14.* Relative $\ell_2$ error at $256 \times 256$ under different refinement modes (all methods use the BIG configuration). Lower is better.

| Refinement mode | Gray–Scott | Allen–Cahn | CNS(1.0, 0.01) | CNS(0.1, 0.01) | SWE | Avg. |
|---|---|---|---|---|---|---|
| No refinement | 2.104 | 1.313 | 1.299 | 0.245 | 0.408 | 1.086 |
| Ours (AMR 25%) | 2.010 | 1.133 | 1.174 | 0.244 | 0.250 | 0.962 |
| Full refinement | 1.687 | 1.012 | 1.017 | 0.204 | 0.282 | 0.840 |

Table 14 reports the $256 \times 256$ results on five PDE benchmarks: Gray–Scott, Allen–Cahn, CNS(1.0, 0.01), CNS(0.1, 0.01), and SWE. Across all equations, our AMR-based refinement reduces error compared with the no-refinement baseline, and the average error decreases from $1.086$ to $0.962$. For instance, on Gray–Scott and Allen–Cahn, refinement improves the error from $2.104$ to $2.010$ and from $1.313$ to $1.133$, respectively. On CNS(1.0, 0.01), the error decreases from $1.299$ to $1.174$, while on SWE the error reduces from $0.408$ to $0.250$. For CNS(0.1, 0.01), the improvement is marginal ($0.245$ vs. $0.244$), suggesting that this setting may be less sensitive to refinement under the same budget.

Full refinement attains the lowest average error ($0.840$), as expected from allocating the highest uniform resolution across the domain. Nevertheless, AMR with a 25% refinement ratio provides a favorable trade-off: it achieves a substantial fraction of the accuracy gain of *Full refinement* while refining only a subset of patches. Overall, these results indicate that refinement-aware tokenization transfers reliably to higher spatial resolutions under the same Transformer backbone capacity.

We also evaluate the same setting at $512 \times 512$ using bilinearly upsampled inputs. The purpose of this experiment is to test whether the refinement mechanism remains beneficial at a higher spatial resolution under the same controlled protocol, rather than to claim a fully optimized high-resolution training setup.

*Table 15.* Relative $\ell_2$ error at $512 \times 512$ using bilinearly upsampled data under the BIG configuration. Lower is better.

| Refinement mode | Gray–Scott | Allen–Cahn | CNS(1.0, 0.01) | CNS(0.1, 0.01) | SWE |
|---|---|---|---|---|---|
| No refinement | 2.430 | 1.524 | 1.270 | 0.255 | 0.457 |
| Ours (AMR 25%) | 2.190 | 1.323 | 1.146 | 0.235 | 0.258 |
| Full refinement | 1.681 | 1.098 | 1.042 | 0.212 | 0.283 |

Table 15 shows that AMR continues to improve over no refinement at $512 \times 512$ on all five benchmarks. However, full refinement achieves lower error on most equations, which is expected because it allocates the finest tokenization uniformly across the entire domain. This controlled setup uses bilinear upsampling from the original $128 \times 128$ fields, which can smooth local activity patterns and weaken the signal used by the activity-based indicator. Therefore, we interpret the $512 \times 512$ results as evidence that AMR remains beneficial over coarse tokenization at higher resolution, while also showing that high-resolution refinement and indicator design remain important directions for future work.

### F.5. Formal Definitions of Positional Encoding Schemes

Let $\mathcal{P}_c = \{1, \ldots, N_c\}$ denote the index set of coarse patches (tokens) on the standard uniform grid, and let $\mathcal{P}_f$ denote the index set of refined tokens produced by quadtree subdivision. Each refined token corresponds to a pair $(i, q)$, where $i \in \mathcal{P}_c$

is its parent coarse patch and $q \in \{1, 2, 3, 4\}$ denotes the quadrant (leaf id) within the refined coarse patch.

For each token $\tau$ (either coarse or refined), we associate its patch-center coordinate $\mathbf{p}_\tau = (x_\tau, y_\tau) \in [0, 1]^2$ and its refinement depth $d_\tau \in \{0, 1\}$ (0 for coarse, 1 for refined). Let $\mathbf{z}_\tau \in \mathbb{R}^D$ be the input token embedding produced by the patch encoder (before adding positional information).

**(1) Learnable PE with Quadtree Offsets**   We introduce a learnable embedding table for coarse patches,

$$\mathbf{E}_\mathrm{c} \in \mathbb{R}^{N_\mathrm{c} \times D}, \qquad \mathbf{e}_i^\mathrm{c} = \mathbf{E}_\mathrm{c}[i] \in \mathbb{R}^D,$$

and a learnable embedding table for quadtree leaf offsets,

$$\mathbf{E}_\mathrm{leaf} \in \mathbb{R}^{4 \times D}, \qquad \mathbf{e}_q^\mathrm{leaf} = \mathbf{E}_\mathrm{leaf}[q] \in \mathbb{R}^D.$$

The positional embedding is defined as

$$\mathbf{e}_\tau = \begin{cases} \mathbf{e}_i^\mathrm{c}, & \tau = i \in \mathcal{P}_\mathrm{c}, \\ \mathbf{e}_i^\mathrm{c} + \mathbf{e}_q^\mathrm{leaf}, & \tau = (i, q) \in \mathcal{P}_\mathrm{f}. \end{cases}$$

Finally, we inject it by simple addition at the Transformer input:

$$\mathbf{h}_\tau^0 = \mathbf{z}_\tau + \mathbf{e}_\tau, \qquad \forall \tau \in \mathcal{P}_\mathrm{c} \cup \mathcal{P}_\mathrm{f}.$$

This construction distinguishes coarse vs. refined tokens but does not explicitly encode continuous geometry.

**(2) Sinusoidal PE over (Pos + Depth) with Dimension Splitting.**   We define a fixed sinusoidal feature map $\Phi_\mathrm{sin}(\cdot\, ; D')$ : $\mathbb{R} \to \mathbb{R}^{D'}$ with frequencies $\{\omega_m\}_{m=0}^{\lfloor D'/2 \rfloor - 1}$:

$$\Phi_\mathrm{sin}(s; D') = \big[\sin(\omega_0 s), \cos(\omega_0 s), \ldots, \sin(\omega_{M-1} s), \cos(\omega_{M-1} s)\big], \quad M = \left\lfloor \frac{D'}{2} \right\rfloor.$$

Given the embedding dimension $D$, we allocate

$$D_x = \left\lfloor \frac{D}{3} \right\rfloor, \qquad D_y = \left\lfloor \frac{D}{3} \right\rfloor, \qquad D_d = D - D_x - D_y.$$

The final positional encoding is formed by concatenation:

$$\mathbf{e}_\tau = \Big[\Phi_\mathrm{sin}(x_\tau; D_x) \,\|\, \Phi_\mathrm{sin}(y_\tau; D_y) \,\|\, \Phi_\mathrm{sin}(d_\tau; D_d)\Big] \in \mathbb{R}^D,$$

and injected additively:

$$\mathbf{h}_\tau^0 = \mathbf{z}_\tau + \mathbf{e}_\tau.$$

This scheme explicitly encodes continuous coordinates and depth, but its functional form is fixed.

**(3) FiLM PE (Pos + Depth) via Learned Conditioning**   We first embed position and depth into two half-dimensional vectors:

$$\mathbf{u}_\tau^\mathrm{pos} = \phi_\mathrm{pos}(x_\tau, y_\tau) \in \mathbb{R}^{D/2}, \qquad \mathbf{u}_\tau^\mathrm{dep} = \phi_\mathrm{dep}(d_\tau) \in \mathbb{R}^{D/2},$$

where $\phi_\mathrm{pos}$ and $\phi_\mathrm{dep}$ are lightweight MLPs. We concatenate them into a conditioning vector

$$\mathbf{g}_\tau = \big[\mathbf{u}_\tau^\mathrm{pos} \,\|\, \mathbf{u}_\tau^\mathrm{dep}\big] \in \mathbb{R}^D.$$

A learnable projection $\psi$ then predicts FiLM parameters (scale and shift),

$$(\boldsymbol{\gamma}_\tau, \boldsymbol{\beta}_\tau) = \psi(\mathbf{g}_\tau), \qquad \boldsymbol{\gamma}_\tau, \boldsymbol{\beta}_\tau \in \mathbb{R}^D.$$

Finally, instead of additive injection, we modulate the input token embedding via

$$\mathbf{h}_\tau^0 = \mathrm{FiLM}(\mathbf{z}_\tau; \boldsymbol{\gamma}_\tau, \boldsymbol{\beta}_\tau) = (1 + \boldsymbol{\gamma}_\tau) \odot \mathbf{z}_\tau + \boldsymbol{\beta}_\tau.$$

Compared to additive positional embeddings, FiLM allows multiplicative, token-wise, scale-aware conditioning, which is particularly suitable for heterogeneous multi-scale token layouts.

*Table 16.* Averaged relative $\ell_2$ error (lower is better) under different positional encoding designs.

| Horizon | Positional encoding | | |
|---|---|---|---|
| | Learnable PE | Sinusoidal PE (pos+depth) | FiLM PE (pos+depth) |
| 1-step | 1.080 | 1.006 | 0.972 |
| 10-step | 2.457 | 2.387 | 2.261 |

**Results and Analysis**   Table 16 shows a clear and consistent ranking among the three positional encoding strategies. Overall, incorporating explicit *(pos+depth)* signals is crucial for AMR token sets: both sinusoidal and FiLM variants substantially outperform the purely learnable embedding design, and the gap becomes more pronounced under long-horizon autoregressive rollout.

**Learnable PE performs the worst.** The learnable baseline yields the highest error (1.080 for 1-step and 2.457 for 10-step), suggesting that naive trainable embeddings provide insufficient inductive bias for multi-scale token layouts. Although the quadtree leaf offsets allow refined tokens to be distinguished from coarse tokens, the resulting embedding still lacks *metric structure*: token coordinates, neighborhood relations, and cross-scale alignments are not encoded explicitly. Consequently, the Transformer must infer how refined tokens relate to their parent patch and how refinement interacts with spatial locality *solely from data*. This is particularly challenging in the foundation-model setting where training spans diverse PDE families and regimes; a purely learnable table must simultaneously memorize many geometry patterns, which can lead to under-generalization or brittle behavior when the refinement pattern shifts across samples. This weakness is amplified in 10-step rollout: small geometric inconsistencies at the token level can translate into systematic local errors (e.g., misaligned gradients or smeared interfaces), which then propagate and compound through autoregressive feedback.

**Sinusoidal PE improves accuracy but remains rigid.** Replacing learnable embeddings with fixed sinusoidal features over $(x, y, d)$ improves performance to 1.006 (1-step) and 2.387 (10-step). This gain indicates that injecting an explicit continuous geometry prior is beneficial: the network receives consistent signals about spatial location and refinement depth across all samples, alleviating the burden of learning geometry from scratch. However, sinusoidal encodings impose a predetermined functional form and treat the three coordinates in a largely separable manner (via concatenation over $x$, $y$, and $d$). In AMR settings, the effective notion of locality is *scale-dependent*: refined tokens represent smaller regions and should interact differently with neighbors compared to coarse tokens. A rigid sinusoidal basis may be overly constrained for capturing such hierarchical interactions, especially when the optimal representation requires nonlinear coupling between position and depth (e.g., depth-dependent receptive fields or scale-dependent attention biases). As a result, sinusoidal PE provides a strong but somewhat inflexible prior, leading to improvements over learnable PE but falling short of the best design.

**FiLM PE achieves the best short- and long-horizon performance.** Our FiLM-based design consistently performs the best (0.972 for 1-step and 2.261 for 10-step). Compared with additive PE, FiLM introduces multiplicative conditioning, effectively allowing the model to *reparameterize* token features as a function of $(x, y, d)$. This is particularly important for multi-scale token sets: depth information can modulate the representation to reflect scale-dependent semantics (e.g., refined tokens emphasizing high-frequency content), while spatial coordinates provide stable alignment across samples. Empirically, the improvement is more evident under 10-step rollout, which highlights that FiLM not only improves one-step fitting, but also enhances long-horizon stability by reducing geometric mismatches that would otherwise accumulate through autoregressive inference.

**Takeaway.** These results support the hypothesis that AMR-aware Transformers require positional encodings that are both *geometrically grounded* (explicit $(x, y, d)$) and *expressive* (learnable nonlinear modulation). Purely learnable tables lack the necessary structural bias, whereas fixed sinusoidal features are partially constrained by their rigid basis. FiLM conditioning offers a favorable middle ground: it preserves strong geometric priors while enabling flexible, depth-dependent feature adaptation, yielding the most robust accuracy–stability trade-off. Unless otherwise stated, we adopt FiLM PE (pos+depth) in all subsequent experiments.

## F.6. Input Noise Magnitude

We investigate the effect of injecting Gaussian noise into the input as an implicit regularizer, and evaluate the averaged relative $\ell_2$ error under both one-step prediction and long-horizon autoregressive rollout. As summarized in Table 17, a moderate noise level ($\sigma = 0.01$) yields the best overall performance, achieving the lowest error for both 1-step prediction (0.972) and 10-step rollout (2.261).

*Table 17.* Ablation on the magnitude of injected Gaussian noise (lower is better). We report averaged relative $\ell_2$ error for one-step prediction and 10-step autoregressive rollout. A moderate noise level (0.01) consistently achieves the best performance on both short-term and long-horizon forecasting.

| Horizon | Noise $= 0$ | Noise $= 0.001$ | Noise $= 0.01$ | Noise $= 0.02$ |
|---------|-------------|-----------------|----------------|----------------|
| 1-step  | 1.125       | 1.125           | **0.972**      | 1.496          |
| 10-step | 3.315       | 3.478           | **2.261**      | 3.504          |

This consistent improvement suggests that moderate perturbations encourage the model to learn a smoother and more robust dynamics mapping, effectively reducing sensitivity to small input variations and mitigating exposure bias during autoregressive inference. In contrast, the noiseless setting ($\sigma = 0$) and a very small noise ($\sigma = 0.001$) provide insufficient regularization, resulting in noticeably larger rollout errors due to error accumulation. On the other hand, excessive noise ($\sigma = 0.02$) starts to distort physically meaningful local structures and degrades both short-term accuracy and long-term stability. Therefore, we adopt $\sigma = 0.01$ as the default noise magnitude in all subsequent experiments.

