# OpenReview forum: "MeshTok: Efficient Multi-Scale Tokenization for Scalable PDE Transformers"
_ICML.cc/2026/Conference — ICML 2026 regular_

### Official Review · Reviewer_c2Jp · 2026-02-28

**Soundness:** 3
**Presentation:** 3
**Significance:** 3
**Originality:** 2
**Overall Recommendation:** 5
**Confidence:** 3

**Summary:**

This paper introduces a new autoregressive foundation model for PDEs. The focus of the transformer-based architecture is an adaptive mesh refinement technique. The algorithm makes a decision to keep the low-resolution patches or to subdivide them to a lower resolution. This decision is made, for example, according to the gradients in the patch. A positional encoding is added that considers the resolution level. After the encoding stage, a transformer module is applied, which compute the attention between all spatial points of the current timestep and  tokens from previous steps. The model is applied autoregressively. In the experiments, the author show that the model can surpass several other PDE foundation models.

**Compliance With Llm Reviewing Policy:**

Affirmed.

**Final Justification:**

The authors responded well to my initial concerns and provided further results. Therefore, I decided to increase my score to accept.

**Key Questions For Authors:**

1. Did you also compare the inference time to the other baseline models?

**Limitations:**

yes

**Strengths And Weaknesses:**

Strengths:
1. The refinement strategy makes the model more efficient than the standard transformer. The model is more accurate than the baseline foundation models.
2. The experimental comparisons to the other foundation models seem to be very well planned. The ablations in the appendix carefully look at other token merging approaches.

Weaknesses:
1. The method only decides between two levels of resolution.
2. The term "indicator network" remains a bit unclear in the main paper. In 3.1, the activity-based indicator is introduced. In line 212, an "indicator network" is introduced. At this point, it remains unclear that the activity-based indicator is supposed to be a type of indicator "network". Another name such as "module" would be more clear and it would also make sense to reiterate here that the activity-based sampling is indeed a version of this "network".
3. The model was not compared to the baselines in the fine-tuning experiments.

---

> ### Author Rebuttal · Authors · 2026-03-31
>
> ## Response to Reviewer
>
> We thank the reviewer for the helpful comments and questions. We address them below.
>
> ### 1. Two-level refinement and the indicator terminology
>
> We agree that the current paper only studies a **two-level** refinement scheme. This is a deliberate design choice: we use the simplest coarse/fine setting to isolate the effect of adaptive token allocation and to keep the token-budget and MAC accounting clean. In 2D, each additional dyadic refinement level increases the local token count by a factor of 4; equivalently, two additional refinement steps already correspond to **16×** as many local tokens as the coarsest representation. This quickly raises the computational cost and makes budget-matched comparison much less clean. We therefore focus on the simplest setting that already captures the main benefit of adaptive token allocation, while leaving deeper hierarchies as a meaningful future direction. The framework itself is not conceptually restricted to two levels.
>
> We also agree that the term **“indicator network”** is not the clearest wording. In our method, this component should be understood more generally as an **indicator/module**, which can be instantiated in different ways. The proposed **activity-based indicator** is one such instantiation, and in the paper we already compare different indicator designs to study their impact. We will revise the wording to make this relationship explicit and avoid confusion.
>
> ### 2. Fine-tuning comparison with the baseline
>
> We appreciate this suggestion and have added the missing fine-tuning comparison for the baseline. For fairness, the baseline uses the same setup as our model throughout both pretraining and fine-tuning, including the data setting, pretraining compute budget, fine-tuning compute budget, and training protocol. The new results are shown below.
>
> | Model | Setting | Shear flow | Burgers | Black-Scholes-Barenblatt | React-Diff |
> |---|---|---:|---:|---:|---:|
> | Baseline | Scratch | 4.827 | 0.435 | **0.123** | 20.626 |
> | Baseline | Pretrained -> finetune | 3.487 | 0.277 | 0.193 | 9.414 |
> | Ours | Scratch | 3.817 | 0.324 | 0.127 | 11.096 |
> | Ours | Pretrained -> finetune | **3.334** | **0.252** | 0.142 | **8.106** |
>
> These results show that pretraining improves performance on most downstream tasks for both methods, and that our model achieves better finetuning performance on **3 out of 4** downstream tasks. We will include this comparison in the revision.
>
> ### 3. Inference time and compute-matched trade-off
>
> We also add an inference-time comparison against baseline models, as requested. For fairness, all timing results are measured under identical settings, including the same GPU environment, the same use of AMP, and the same inference protocol, and each model is run **50 times** with the **mean runtime (ms)** reported.
>
> | Metric | DeepONet | FNO | ViT | MPP | DPOT | MoE-POT | BCAT | Ours |
> |---|---:|---:|---:|---:|---:|---:|---:|---:|
> | Runtime (ms) | 44.78 | 30.63 | 7.40 | 90.41 | 17.11 | 50.03 | 17.79 | 39.74 |
>
> Beyond raw runtime, we further performed a **compute-matched** comparison to better evaluate the accuracy-efficiency trade-off. Specifically, we increased the width/depth of the **no-refinement** model and reduced the width/depth of the **full-refinement** model so that their MACs are close to that of our AMR model. For fairness, these baselines use the same data setting as our model. The results are:
>
> | Model | N/step | Arch. (L,d,f;p) | MACs | Params (M) | Rel. ↓ |
> |---|---:|---|---:|---:|---:|
> | Uniform (coarse), width increased | 64 | (8,640,1800;8) | 2.650e9 | 47.19 | 0.010294 |
> | Uniform (coarse), depth increased | 64 | (14,512,1280;8) | 2.760e9 | 47.32 | 0.011067 |
> | **AMR (activity_based)** | **112** | **(8,512,1280;8)** | **2.804e9** | **31.43** | **0.009723** |
> | Uniform (full refine), width reduced | 256 | (8,320,800;16) | 2.747e9 | 10.38 | 0.010777 |
> | Uniform (full refine), depth reduced | 256 | (4,512,1280;16) | 3.355e9 | 13.67 | 0.010632 |
>
> Even under matched MACs, AMR achieves the lowest error. This suggests that the gain is not solely explained by token count or backbone compute under this matched accounting, but also by more effective allocation of computation.
>
> ### Summary
>
> In the revision, we will:
> 1. clarify that the current paper studies a two-level coarse/fine instantiation and discuss multi-level refinement as future work;
> 2. replace the ambiguous term **“indicator network”** with clearer wording such as **indicator** or **indicator module**, and explicitly state that the activity-based indicator is one instantiation;
> 3. add the missing **baseline fine-tuning** comparison;
> 4. add the **inference-time** comparison and the **compute-matched** analysis.
>
> We thank the reviewer again for these helpful suggestions.

---

> > ### Author Rebuttal · Reviewer_c2Jp · 2026-04-02
> >
> > Thank you for your response. I have two further questions:
> >
> > 1. You only wrote "baseline" in your rebuttal for 2 - which baseline is it?
> > 2. For Table 1 in the paper, was your model also pretrained? The caption seems to suggest so ("We evaluate pretrained models on multiple datasets "), but judging from the context, it does not seem to be the case. Was MPP or DPOT also pretrained here?

---

> > > ### Author Response · Authors · 2026-04-02
> > >
> > > Thank you for pointing this out. We agree that our previous wording was imprecise, and we will revise it to make the experimental protocol explicit.
> > >
> > > For the first point, the additional fine-tuning baseline in Sec. 2 of the rebuttal is BCAT. In this comparison, BCAT and our method use the same fine-tuning dataset, and both are trained for 5 epochs, with 4000 training iterations per epoch.
> > >
> > > For the second point, in Table 1, all methods — including MPP, DPOT, and our method — are trained from scratch under the same multi-dataset training setup. Specifically, each method is trained for 20 epochs, with 4000 training iterations per epoch, which is sufficient for each model to reach a stable performance regime. All methods are trained on the same collection of datasets, including Gray-Scott from The Well, Allen-Cahn from PDENNEval, and CNS(1.0, 0.01), CNS(0.1, 0.01), and SWE from PDEBench.
> > >
> > > We will revise the wording in the paper accordingly to correct these imprecise statements and make the above protocol explicit.

---

### Official Review · Reviewer_u7bB · 2026-03-12

**Soundness:** 2
**Presentation:** 3
**Significance:** 2
**Originality:** 3
**Overall Recommendation:** 3
**Confidence:** 4

**Summary:**

This paper studies adaptive tokenization for Transformer-based PDE forecasting. Instead of using a uniform patch partition over the spatial domain, the proposed MeshTok framework refines only a subset of coarse patches based on an activity score derived from local gradients/Laplacians, producing a mixed sequence of coarse and fine tokens that is processed by a shared Transformer. The method also introduces a geometry-aware positional encoding that incorporates both token coordinates and scale information. Empirically, the paper evaluates MeshTok on several PDE benchmarks and compares it with PDE foundation-model baselines, reporting improved accuracy over uniform-token baselines and a better accuracy-efficiency trade-off than full refinement. The paper also includes theoretical discussion aimed at motivating why refinement can enlarge the realizable function class relative to a coarse partition.

**Compliance With Llm Reviewing Policy:**

Affirmed.

**Key Questions For Authors:**

1.The paper claims “convergence and approximation guarantees,” but the main theorems appear to establish representational lower bounds and realizable-class inclusion under fairly strong assumptions (e.g., existence of an alignment map and closure assumptions on the encoder/decoder classes). Can the authors clarify exactly what theoretical guarantee they believe is new here, and in what sense it is a convergence result rather than an expressivity argument? A convincing clarification would improve my assessment of the theoretical contribution.
2.In the main method, is refinement during autoregressive rollout recomputed from the model’s predicted states at each step, or is it fixed from the conditioning window / teacher-forced ground truth? This matters because an activity-based policy may be much easier to apply when true states are available than during long-horizon rollout. Clarifying this could affect my view of the practical robustness of the method.
3.Can the authors report variance over multiple random seeds, at least for the main comparisons in Table 1 and the scaling/efficiency study? Some gains over strong baselines are relatively small, and confidence intervals or standard deviations would help determine whether these improvements are robust.
4.The efficiency analysis compares No refinement / Ours / Full refinement within the proposed framework. Can the authors also provide a more direct efficiency-accuracy comparison against at least one strong baseline such as BCAT or DPOT under matched parameter count or matched runtime? This would make the practical value of MeshTok clearer.
5.Figure 1 and parts of the text suggest an indicator network, while Section 3.1 presents a hand-crafted activity indicator and the appendix later discusses learned indicators. Can the authors clarify what exactly is used in the main model reported in Tables 1–4, and why that choice is preferred over the learned alternatives?

**Limitations:**

The paper does discuss a limitation in Section 4.6, but the discussion is fairly brief and focuses mainly on diminishing gains at larger model scales. I think the paper should also explicitly mention at least three additional limitations: (i) the main refinement policy is largely hand-crafted and may depend on PDE-specific inductive bias; (ii) the theory does not directly justify the specific practical method under realistic model classes; and (iii) the current evaluation does not fully establish robustness across seeds or matched-budget comparisons with strong baselines. The societal-risk discussion can remain brief since this is a low-risk scientific ML paper, but the limitations section should be somewhat more complete.

**Strengths And Weaknesses:**

1.The idea of adaptive tokenization for PDE Transformers is intuitive and well motivated. The method allocates finer tokens to dynamically active regions while keeping coarse tokens elsewhere, which is conceptually aligned with adaptive mesh refinement. The empirical evaluation is reasonably broad, including several PDE benchmarks and comparisons with both operator-learning methods and recent PDE foundation models. Results suggest that the approach can improve accuracy over uniform tokenization and achieve a better accuracy–efficiency trade-off than full refinement. However, the theoretical contribution is somewhat limited. The main results appear to establish representational arguments under relatively strong assumptions rather than providing rigorous convergence guarantees for the practical model. As a result, the theory mainly serves as intuition for adaptive refinement rather than a strong justification of the specific architecture.
2.The paper is generally clear and well organized. The motivation and high-level framework are easy to follow, and the figures help illustrate the token refinement mechanism. However, the boundary between the proposed contribution and existing PDE Transformer designs could be clarified more clearly, since much of the backbone architecture appears similar to prior work.
3.The problem is relevant for scaling Transformer-based PDE models. Adaptive token allocation could be useful for improving efficiency when modeling high-resolution PDE dynamics. That said, the contribution seems more like an architectural refinement rather than a fundamentally new modeling paradigm.
4.The work combines AMR-style refinement ideas with Transformer tokenization in a clean way, which is a reasonable design contribution. Nevertheless, the novelty appears moderate, as the main gain comes from integrating existing ideas rather than introducing a fundamentally new learning framework.

---

> ### Author Rebuttal · Authors · 2026-03-31
>
> We thank the reviewer for the constructive feedback.
>
> **Q1. Theory.** We agree that the previous text blurred expressivity lemmas with the main guarantee. The result is not an optimization-convergence theorem for the exact trained practical model. It is a budget-aware approximation/convergence guarantee for the MeshTok hypothesis class under simultaneous partition refinement and budget growth.
>
> Let $S_{\Delta t}:\mathcal M\to Y$ be $L_S$-Lipschitz on a compact solution manifold, let $\eta_P:=\sup_{u\in\mathcal M}\|u-\Pi_Pu\|$, and let $\mathcal F_{P,W,m}$ denote the MeshTok class under partition $P$. If the backbone approximation error satisfies
> $$
> \rho_P(W,m)\le C_{\rm tr}W^{-\beta}+C_{\rm loc}m^{-\gamma},
> $$
> and ${\rm Cost}(P,W,m)\le B$, then
> $$
> \mathcal E_P^\star(B):=\inf_{{\rm Cost}\le B}\inf_{F\in\mathcal F_{P,W,m}}
> \sup_{u\in\mathcal M}\|S_{\Delta t}(u)-F(u)\|
> \le L_S\eta_P+\Psi(N(P),B),
> $$
> with $\Psi(N,B)=C_1(N/B)^\gamma+C_2(N^2/B)^\beta$. Hence $\mathcal E^\star_{P_n}(B_n)\to0$ whenever $\eta_{P_n}\to0$ and $\Psi(N(P_n),B_n)\to0$.
>
> Proof sketch:
> $$
> \|S(u)-F(u)\|\le \|S(u)-S(\Pi_Pu)\|+\|S(\Pi_Pu)-F(u)\|.
> $$
> The first term is controlled by Lipschitz continuity; the second is controlled by the backbone approximation bound under the budget. We will therefore revise the wording from “convergence guarantees” to “approximation/convergence under refinement and budget growth,” and state former Theorems 2--3 as auxiliary.
>
> We will also keep **Theorem 2 (partition approximation on interface states)** because it makes the geometric mechanism explicit: for piecewise-smooth states with a $C^2$ interface, if $H$ denotes the coarse cell size in the bulk and $h$ the fine cell size used near the interface, then uniform partitions incur an $L^2$ projection lower bound $\Omega(H^{1/2})$, whereas AMR achieves $O(H+h^{1/2})$. The key assumption is that the nonsmooth part is confined to a tubular neighborhood of a codimension-one manifold. In $d$ dimensions, this means the refined region is effectively supported on a $(d-1)$-dimensional set rather than the full $d$-dimensional domain, so fine cells are introduced only near the interface instead of everywhere. This explains why targeted refinement is especially effective when the error is localized near interfaces.
>
> **Q2. Rollout refinement.** During autoregressive rollout, refinement is **recomputed from the model’s current predicted state at every step**. It is **not** fixed from teacher-forced ground truth, and it does **not** require any true refinement labels. The activity score depends only on the current field (gradient/Laplacian), so our long-horizon results already include the more practical and harder prediction-dependent refinement setting.
>
> **Q3. Seed robustness.** For a fair comparison, we use exactly the same data and experimental settings as in the main paper, and vary only the random seed to assess robustness. Due to the rebuttal timeline, we ran three independent seeds for the most important scaling study. The conclusion is consistent across all scales: AMR always improves over No refinement, with small variance across runs.
>
> | Scale | No ref. | AMR | Full ref. |
> |---|---:|---:|---:|
> | Small | 2.013±0.031 | 1.631±0.009 | 1.933±0.071 |
> | Big | 1.122±0.015 | 0.977±0.007 | 0.914±0.009 |
> | Large | 0.917±0.007 | 0.842±0.006 | 0.727±0.016 |
>
> These results show that the improvement of AMR is consistent across the three seeds in this scaling study, with small variance across runs.
>
> **Q4. Compute-matched baseline.** We additionally compared against a compute-matched BCAT baseline by adjusting the width/depth so that the MACs are comparable under the same accounting used in our paper. We keep the data setup and training protocol the same as in the main paper. Under this accounting, MeshTok achieves the best relative error.
>
> | Model, Arch. (L,d,f) | MACs | Params(M) | Rel. ↓ |
> |---|---:|---:|---:|
> | patch_size=16, (8,640,1800) | 2.650e9 | 47.20 | 0.010294 |
> | patch_size=16, (14,512,1280) | 2.760e9 | 47.33 | 0.010652 |
> | **AMR** | **2.804e9** | **31.43** | **0.009723** |
> | patch_size=8, (8,320,800) | 2.747e9 | 10.39 | 0.010777 |
> | patch_size=8, (4,512,1280) | 3.355e9 | 13.68 | 0.010632 |
>
> Under this accounting, MeshTok achieves the lowest relative error among the compared models.
>
> **Q5. Indicator used in the main model.** Unless otherwise stated, the main experiments (Tables 1--4) use the **activity-based** indicator. Figure 1 presents a generic indicator module, whereas the main model reported in the paper uses a hand-crafted activity score, as it is label-free, computationally inexpensive, and empirically effective. In the appendix, we compare random, activity-based, and a posteriori indicators. The results show that the activity-based indicator achieves the best overall accuracy–efficiency trade-off, while learned and a posteriori variants incur noticeable additional cost. We will clarify this distinction explicitly in the main text.

---

### Official Review · Reviewer_sKtY · 2026-03-16

**Soundness:** 3
**Presentation:** 3
**Significance:** 3
**Originality:** 3
**Overall Recommendation:** 4
**Confidence:** 3

**Summary:**

In this paper, an adaptive multi-scale tokenization framework named MeshTok for Transformer-based PDE modeling is introduced. The main idea is inspired by adaptive mesh refinement (AMR): instead of using uniform spatial patches (as in standard ViT-style tokenization), the method selectively refines regions of the spatial domain that exhibit higher activity or complexity. The refinement is guided by an activity indicator based on gradient magnitude and Laplacian energy computed from the PDE field. Refined patches are subdivided into smaller tokens while smooth regions remain represented by coarse tokens. These heterogeneous tokens are then processed jointly by a standard Transformer backbone using a unified token sequence with a geometry-aware positional encoding that includes both spatial coordinates and token scale. The experiment results show that the proposed method can improve the efficiency–accuracy trade-off on PDE benchmarks.

**Compliance With Llm Reviewing Policy:**

Affirmed.

**Key Questions For Authors:**

- The refinement indicator selects patches independently at each timestep, which means the token layout changes over time. What is the impact/trade-off of this change comparing to uniform token setup? Would it affect the temporal consistency?
- Is it possible for this method to extend to unstructured mesh scenarios?
- How would the method perform on higher spatial resolutions cases?
- How is the method sensitive to the refinement indicator design

**Limitations:**

yes

**Strengths And Weaknesses:**

### Strengths
- The method is conceptually grounded in adaptive mesh refinement, a well-established technique in numerical PDE solvers. The connection between AMR and token allocation looks reasonable.
- The authors provide theoretical analysis demonstrating that adaptive refinement increases the realizable function class compared to coarse tokenization, along with bounds related to token collisions and approximation error.
- The experimental evaluations are relatively comprehensive, including multiple datasets covering different PDE families, baselines neural operators (FNO, DeepONet) and Transformer-based models, ablation studies on refinement strategies, positional encodings, noise augmentation, and refinement ratios.

### Weaknesses
- The experiments are limited to 2D (256 x 256), it is not clear how the method would benefit in larger or 3D cases.
- The refinement strategy relies on a simple gradient + Laplacian activity score to detect complex regions, without taking the actual PDE solution errors into consideration
- The method is limited to the structured mesh, but in actual scenarios, unstructed meshes are better representations

---

> ### Author Rebuttal · Authors · 2026-03-31
>
> Thank you for these helpful comments and questions. We are happy to clarify the scope and design choices of our method.
>
> **(1) Time-varying refinement and temporal consistency.** For time-dependent PDEs, the spatial regions that require finer resolution typically evolve over time. For this reason, we allow the refinement pattern to change across timesteps, rather than enforcing a single fixed token layout for the entire trajectory. We believe this is a more natural design for evolutionary PDEs, since a fixed layout may over-allocate fine tokens to regions that have become smooth while under-resolving newly active regions. Empirically, this dynamic design is beneficial: with all other settings kept identical and only the refined-token selection changed, the average error of a fixed-layout variant is 1.068, while the proposed activity-based refinement achieves 0.972, showing that adapting the partition over time improves prediction quality. Importantly, although the refined locations change, the **refinement ratio** is predefined, so the number of refined patches at each timestep is fixed. As a result, the total number of tokens entering the Transformer remains the same across timesteps, and therefore the compute, remain essentially unchanged.
>
> **(2) Extension to unstructured meshes.** Our current architecture follows a ViT-style patch tokenization pipeline, whose initial tokenization step is naturally defined on structured grids. Extending the same idea to unstructured meshes is an interesting direction, but it would usually require a different backbone, e.g., graph- or mesh-based neural architectures. We agree that unstructured meshes are important in many practical scenarios. That said, systematically studying this setting would substantially broaden the scope of the paper, especially since the scaling behavior of graph-based models is itself a nontrivial issue. We will clarify in the revision that this paper focuses on structured-grid PDE modeling, while extension to unstructured meshes is a meaningful future direction.
>
> **(3) Higher spatial resolutions.** Our main experiments are conducted at **$128 \times 128$**, and we have already extended the study to **$256 \times 256$** in the appendix. We further test **$512 \times 512$** by bilinearly upsampling the data and applying the same pipeline.
>
> **Relative $\ell_2$ error at $512 \times 512$ (BIG configuration; lower is better).**
>
> | Refinement mode | Gray--Scott | Allen--Cahn | CNS(1.0, 0.01) | CNS(0.1, 0.01) | SWE |
> |---|---:|---:|---:|---:|---:|
> | No refinement | 2.430 | 1.524 | 1.270 | 0.255 | 0.457 |
> | Ours (AMR 25%) | 2.190 | 1.323 | 1.146 | 0.235 | 0.258 |
> | Full refinement | 1.681 | 1.098 | 1.042 | 0.212 | 0.283 |
>
> On an upsampled 512×512 setting, AMR still improves over no refinement, while not showing a clear improvement over full refinement. We believe this is mainly due to the current controlled setup: for consistency, we still start from **$8 \times 8 = 64$** coarse patches, which may already be too coarse for **$512 \times 512$**; moreover, the **bilinear upsampling** makes the fields smoother, which weakens the activity signal and makes the activity-based indicator less effective. We will clarify this point in the revision.
>
> **(4) Sensitivity to the refinement indicator.** We agree that the refinement indicator matters, which is precisely why we adopt an activity-based design. Its advantage is that it is both cheap to compute and effective in practice, leading to a strong accuracy-efficiency trade-off. In the paper, we already compare multiple refinement strategies, including a network-based posterior-error-style indicator, and the proposed activity-based indicator consistently achieves a better trade-off overall. We do not use the true solution error itself as an indicator, because it is unavailable at inference time and would amount to an oracle signal. Instead, our goal is to use a causal proxy computable from the current state. Exploring more principled causal proxies, such as residual- or uncertainty-based indicators, is an interesting direction for future work.
>
> Overall, we will revise the paper to better explain: (i) why time-varying refinement is natural for evolutionary PDEs; (ii) why changing layouts do not change token count or inference cost under a fixed refinement ratio; (iii) why the current paper focuses on structured grids; and (iv) why the indicator design is important for the final trade-off.

---

> > ### Author Rebuttal · Reviewer_sKtY · 2026-04-03
> >
> > Thanks for the reply. Though it is still unclear how the method will perform on 3D or more complex, it is not easily to address in a short rebuttal. Please add the discussion you mentioned in the "overall" to the revised paper. In summary, I will keep my scores unchanged.

---

> > > ### Author Response · Authors · 2026-04-08
> > >
> > > Additional 3D experiment on The Well (MEXWELL).
> > > We thank the reviewer for asking about the behavior in 3D. We therefore conducted an additional experiment on the MEXWELL dataset from The Well, using 100 trajectories with 4 physical variables in total: density and the three velocity components. Since this 3D set is much smaller than our main training sets, we slightly adjusted the training schedule to avoid severe overfitting: all models are trained for 5 epochs, with 1000 iterations per epoch.
> > >
> > > For the 3D setting, we also slightly adjust the refinement ratio. In 2D, one refined token is replaced by 4 finer tokens, while in 3D it is replaced by 8 finer tokens. To better match this geometric growth of tokens in 3D, we increase the refinement ratio from 0.25 in 2D to 0.375 in 3D. Under this setting, the AMR model has sequence length 232, compared with 512 for full refinement. Since self-attention scales quadratically with sequence length, its attention cost is approximately
> > >
> > > $$
> > > (232 / 512)^2 \approx 20.5\%
> > > $$
> > >
> > > of full refinement.
> > >
> > > The results are summarized below.
> > >
> > > | Metric | No refinement (`patch_num=4`) | AMR (`patch_num=4`, `refine_ratio=0.375`) | Full refinement (`patch_num=8`) |
> > > |---|---:|---:|---:|
> > > | Relative error ↓ | 0.39647 | 0.35851 | 0.29626 |
> > > | Runtime (ms) ↓ | 62.17 | 149.91 | 451.67 |
> > >
> > > These results show a consistent trend in 3D. First, AMR already improves substantially over no refinement, reducing the relative error from 0.39647 to 0.35851, which suggests that the adaptive refinement mechanism remains effective beyond 2D. Second, although full refinement achieves the lowest error in this particular small-data regime, it does so at a much higher computational cost. In contrast, AMR uses less than half of the tokens of full refinement, only about one-fifth of its attention cost, and is about 3.0× faster in runtime, while still obtaining a noticeably better error than the coarse no-refinement model.
> > >
> > > Overall, these results suggest that the proposed method remains effective in 3D, providing a consistent improvement over no refinement while retaining a favorable accuracy-efficiency trade-off. We will add this discussion to the revised paper.

---

### Decision · Program_Chairs · 2026-04-30

**Decision:**

Accept (regular)

**Comment:**

This paper makes a contribution to PDE modeling with Transformers. It proposes an adaptive tokenization scheme that places finer tokens in regions with more complex dynamics and coarse tokens elsewhere, a sensible way to use computation more effectively. The method is well motivated by adaptive mesh refinement, yet simple enough to fit naturally into a standard Transformer pipeline. The experiments are convincing, covering multiple datasets, comparisons to strong baselines, fine-tuning, scaling, and efficiency. Across these settings, the paper shows a consistent improvement in the accuracy-efficiency trade-off, and the results suggest that the gains are not just from adding more computation but from allocating it more effectively.

The main concerns raised in the reviews do not outweigh the work's strengths. The current study focuses on structured grids, and the original submission was mostly centered on 2D settings. The rebuttal addresses these points in a reasonable way and adds supporting evidence that the method remains useful in 3D. The theoretical claims should be read more as approximations and expressive support than as a full convergence result for the trained model, but this does not weaken the paper's practical contribution. Overall, the work is solid, clearly written, and likely to be useful to others working on scientific machine learning and neural PDE solvers. I recommend acceptance.